# LEARN HYBRID PROTOTYPES FOR MULTIVARIATE TIME SERIES ANOMALY DETECTION

**Ke-Yuan Shen**
Hebei Key Laboratory of Machine Learning and Computational Intelligence
College of Mathematics and Information Science
Hebei University Baoding,Hebei,China
`licg@hbu.edu.cn`

## ABSTRACT

In multivariate time series anomaly detection (MTSAD), reconstruction-based models reconstruct testing series with learned knowledge of only normal series and identify anomalies with higher reconstruction errors. In practice, over-generalization often occurs with unexpectedly well reconstruction of anomalies. Although memory banks are employed by reconstruction-based models to fight against over-generalization, these models are only efficient to detect point anomalies since they learn normal prototypes from time points, leaving interval anomalies and periodical anomalies to be discovered. To settle this problem, this paper propose a hybrid prototypes learning model for MTSAD based on reconstruction, named as H-PAD. First, normal prototypes are learned from different sizes of the patches for time series to discover interval anomalies. These prototypes in different sizes are integrated together to reconstruct query series so that any anomalies would be smoothed off and high reconstruction errors are produced. Furthermore, period prototypes are learned to discover periodical anomalies. One period prototype is memorized for one variable of the query series. Finally, extensive experiments on five benchmark datasets show the effectiveness of H-PAD.

## 1 INTRODUCTION

Anomaly detection in multivariate time series is a common but important issue in many fields such as equipment monitoring, healthcare systems and aerospace engineering. Since labeling is time-consuming and labor-intensive, multivariate time series anomaly detection (MTSAD) is regarded as an unsupervised learning task(Eldele et al., 2021). Generally speaking, MTSAD learns knowledge directly from a set of normal data, and detects anomalies with learned normal knowledge.

The most popular MTSAD methods are developed based on reconstruction of time series. In the training phase, a set of time series of only normal points are featured in latent space by an encoder. Following with a decoder, the training set of time series are expected to reconstructed with least losses. In the inference phase, the trained encoder and decoder try to reconstruct a new time series and anomaly points would be discovered. However, the best reconstruction of training time series raises the problem of over-generalization for testing time series, shown as in Figure 1(a). That is, not only normal points in time series are reconstructed excellently, but abnormal points are also reconstructed very well. As a result, abnormal points can not be discovered because they can not be identified with high reconstruction errors no longer. To fight against over-generalization, MEMTO employs a memory bank of normal point prototypes to help reconstruct time series (Song et al., 2024). However, local information should be seriously considered for learning time series since each point is closely related with its neighbours. Moreover, the absence of periodicity in MTSAD lead

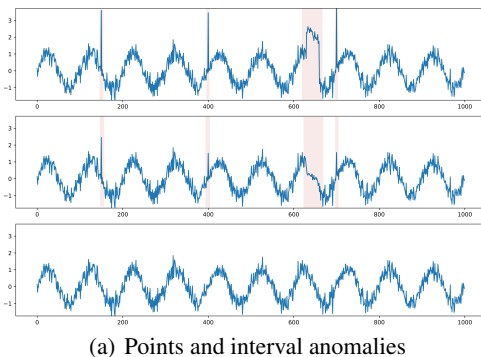 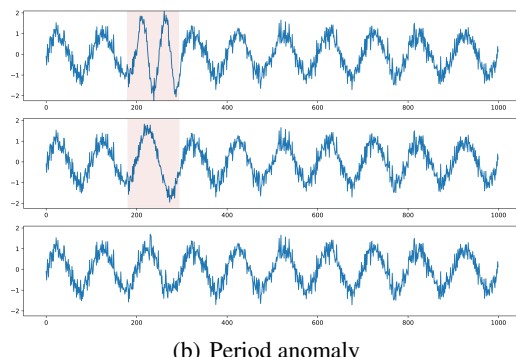

(a) Points and interval anomalies         (b) Period anomaly

Figure 1: Illustrations of over-generalization. The top time series is the time series including anomalies (highlighted in pink). The middle time series is its reconstructed series due to over-generalization that anomalies are reconstructed as well as normal ones. After all, the time series is expected to be reconstructed as the bottom series, immune to anomaly influence.

to the fact that it is difficult to identify long-term anomalies (e.g. period anomalies in Figure 1(b)) only with point prototypes. For detailed description, see appendix A.

This paper proposes an MTSAD model based on learning hybrid prototypes (H-PAD) which consist of patch prototypes in different scales and periodical prototypes (as shown in Figure 2). First of all, H-PAD is designed to learn memory prototypes of the patches in different scales, instead of prototypes of the time points. With patches prototypes, local information in time series is taken into consideration for future reconstruction, which enables the model to identify interval anomalies. And occasional point anomalies can not be reconstructed well with their local information, further preventing the occurrence of over-generalization. Moreover, taking periodicity of time series into account, H-PAD also learns and memorizes period prototypes for time series in multiple variables, one period prototype for one variable. It enables the model to identify long-term anomalies because period prototypes can help to reconstruct the testing time series as a normal series which deviates greatly from input series. Experimental results on five benchmarks illustrate the effectiveness of H-PAD.

The contributions of this paper are as follows:

- We propose a novel framework to learn hybrid prototypes for multivariate time series anomaly detection. Patch prototypes involves local information and period prototypes contains global information. The combination of the patch and period prototypes can well discover point anomalies and interval anomalies as well as period anomalies.

- The model comprehensively considers local information and global information, uses patches to learn local features, and uses periods to learn global information. The learned local features and period features are helpful for learning hybrid prototypes.

- By comprehensively considering the differences between different patch prototypes and time points, and the differences between period prototypes and period, and constructing anomaly scores based on the distance between the original features and the nearest prototype in the feature space, anomalies in the test data can be identified more accurately.

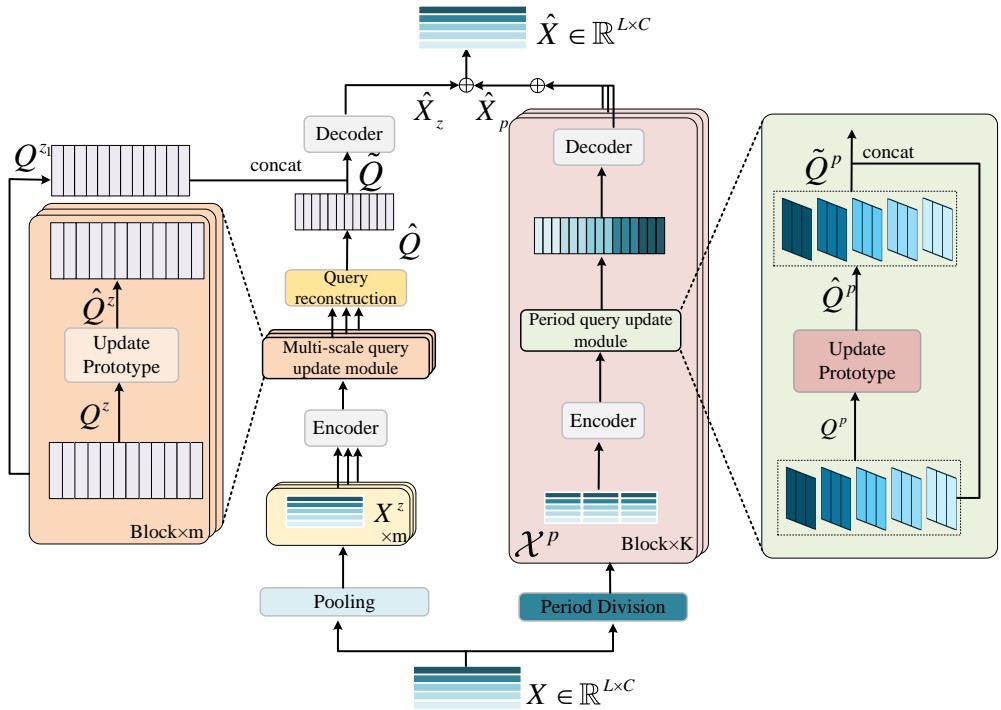

Figure 2: The overall architecture of the H-PAD model.

## 2 RELATED WORKS

Early anomaly detection methods are generally some classical machine learning methods. Machine learning methods include classification-based, density-based and clustering-based methods. Classification-based methods such as decision trees (Liu et al., 2008), support vector machines, one-class SVM (Schölkopf et al., 2001) and support vector data description (Tax & Duin, 2004) were widely used in the field of anomaly detection in the early days. Density-based methods such as local outlier factor (Breunig et al., 2000) and connectivity outlier factor (Tang et al., 2002) calculate local density and local connectivity respectively to determine outliers. In recent years, methods combining density estimation with deep learning, such as DAGMM (Zong et al., 2018) and MPPCACD (Yairi et al., 2017), have also been proposed. Common clustering methods include k-means (Kant & Mahajan, 2019), THOC (Shen et al., 2020), and ITAD (Shin et al., 2020).

Deep learning methods have become the mainstream method for anomaly detection because they can capture complex nonlinear relationships and time dependencies in time series data. The most commonly used method is reconstruction-based. Early methods include LSTM-based encoder-decoder models and LSTM-VAE models (Park et al., 2018). Later, OmniAnomaly (Su et al., 2019) and InterFusion (Li et al., 2021) further extended the LSTM-VAE model. With the deepening of research on reconstruction models, reconstruction models combined with generative models have also been applied to time series anomaly detection, such as BeatGAN (Zhou et al., 2019), a variant of generative adversarial networks. In recent years, Anomaly Transformer (Xu et al., 2022) has introduced the correlation difference between normal points and abnormal points to improve the effect of anomaly detection. Dcdetector (Yang et al., 2023) uses patch learning of local information and permutation-invariant representation based on Anomaly Transformer to improve detection

accuracy. D3R (Wang et al., 2023) addresses drift in time series by dynamically decomposing with data-time mix-attention and externally controlling the reconstruction bottleneck via noise diffusion. DMamba (Chen et al., 2024) proposes a selective state space model with a multi-stage detrending mechanism to enhance long-range dependency modeling and generalization in non-stationary Time Series Anomaly Detection. In addition, to capture the correlation between variables, graph structures are also used in time series anomaly detection. GSC_MAD (Zhang et al., 2024) uses graph structures for anomaly detection and achieves good results.

Memory networks were initially applied in natural language processing (Weston et al., 2014), leveraging reasoning components and long-term memory components to perform inference and learn how to utilize them jointly. The long-term memory is designed to support read and write operations, enabling it to be used for prediction tasks. In the context of question answering (QA), these models were explored where the long-term memory effectively serves as a (dynamic) knowledge base, with the output being a textual response. Later, an improved version of memory networks was proposed (Sukhbaatar et al., 2015), employing end-to-end training and utilizing a recurrent attention model to retrieve memory items. Recently, memory networks have emerged as a powerful tool in various fields, particularly in computer vision Park et al. (2020). For anomaly detection, MemAE Gong et al. (2019) is pioneering in integrating memory networks into an autoencoder framework. Despite its innovativeness, MemAE's performance is limited by the lack of a dedicated mechanism for updating memory prototypes. To overcome this shortcoming, MNAD Park et al. (2020) proposes to update memory prototypes by storing multiple patterns of normal behaviors within the memory framework. While this method marked an improvement over MemAE, it still faces challenges that it is difficult to embrace new information since the memory prototypes are updated with a sum of related queries of fixed weights. As for application in MTSAD, MEMTO Song et al. (2024) provides a solution to address this issue. A gating unit is taken by MEMTO to regulate the amount of new information for prototypes updating during the learning procedure from normal time series.

## 3 PROPOSED METHOD

To begin with, the original time series $\mathbb{X}$ is divided into multiple subsequences of the same length, $\mathbb{X} = \{\mathbf{X}_1, \mathbf{X}_2, \cdots, \mathbf{X}_a\}$. Each subsequence $\mathbf{X} \in \mathbb{X}$ is taken as one time series for training. Let $\mathbf{X} = \{\mathbf{x}_1, \mathbf{x}_2, \cdots, \mathbf{x}_L\}$, where $L$ is the length of the series and $\mathbf{x}_t \in \mathbb{R}^C$ is an observation vector at time $t$. MT-SAD aims at learning knowledge from normal time series and generating labels $\mathbf{Y}_{test} = \{y_1, y_2, \ldots, y_{L_1}\}$ for unseen series $\mathbf{X}_{test} \in \mathbb{R}^{L_1 \times C}$, where $y_t \in \{0, 1\}$, 0 for normal and 1 for abnormal. For reconstruction-based model, anomaly scores are given by $s_t = \|\mathbf{x}_t - \hat{\mathbf{x}}_t\|_2$ where $\hat{\mathbf{x}}_t$ is the reconstructed observation of $\mathbf{x}_t$ with learning knowledge and $\|\cdot\|_2$ is L2-norm. Finally, anomaly labels are determined by the anomaly threshold $\delta$. if $s_t \geq \delta$ then $y_t = 1$, otherwise $y_t = 0$.

### 3.1 LEARNING NORMAL PATCH PROTOTYPES

To learn different scales of temporal information, H-PAD features different sizes of prototypes from different sizes of patches from normal times series. Different scales of local information are contained into different sizes of prototypes and different views of normal features are embedded into patch prototypes.

**Average Pooling.** Given $\mathbf{X} \in \mathbb{R}^{L \times C}$, it is divided into several patches of size $z \in \{1, 2, \cdots, m\}$ without overlapping. All $\mathbf{x}_t$ in the same patch is averaged according to equation 1

$$\mathbf{x}_i^z = \frac{1}{z} \sum_{t=t_0}^{i \cdot z} \mathbf{x}_t \tag{1}$$

where $t_0 = (i - 1) \cdot z + 1$. Thus a new series is generated to be $\mathbf{X}^z = \{\mathbf{x}_1^z, \mathbf{x}_2^z, \cdots, \mathbf{x}_{L_z}^z\}$, where $L_z = \lceil \frac{L}{z} \rceil$. Specifically, $\mathbf{X}^z$ for $z = 1$ is the original time series $\mathbf{X}$ which remains the original information after passing

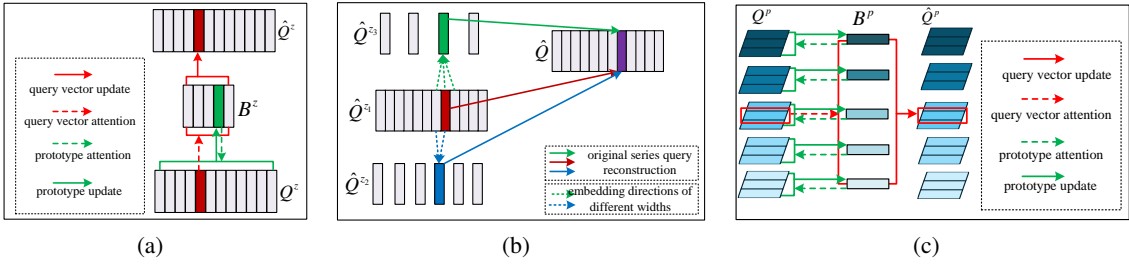

Figure 3: (a) Updating normal patch prototypes. (b) Query reconstruction with normal patch prototypes. (c) Updating period prototypes.

through the pooling layer with a pooling kernel size of 1. With average pooling, different $\mathbf{X}^z$ involve different width local information in order to learn temporal dependencies within different durations.

**Update Patch Prototypes** To learn prototypes in feature space, the generated series $\mathbf{X}^z$ are embedded into the feature space of higher dimension according to the Transformer Encoder

$$\mathbf{Q}^z = Encoder\left(\mathbf{X}^z\right) \tag{2}$$

where $\mathbf{Q}^z = \left\{\mathbf{q}_1^z, \mathbf{q}_2^z, \cdots, \mathbf{q}_{L_z}^z\right\} \in \mathbb{R}^{L_z \times D} \ (D > C)$, which is used to learn patch prototypes in different scales. The detailed information about the Transformer Encoder can be found in Appendix B.

Initializing randomly, patch prototypes $\mathbf{B}^z = \left\{\mathbf{b}_1^z, \mathbf{b}_2^z, \cdots, \mathbf{b}_M^z\right\} \in \mathbb{R}^{M \times D}$ are updated with an update gate $\psi$ based on the query series $\mathbf{Q}^z$ (Figure 3(a)). First of all, the similarity matrix $\mathbf{V}^z = \left(v_{ij}^z\right)_{M \times L_z}$ of patch prototypes $\mathbf{B}^z$ and the query series $\mathbf{Q}^z$ can be derived by

$$v_{ij}^z = \frac{\exp\left(\langle \mathbf{b}_i^z, \mathbf{q}_j^z \rangle / \tau\right)}{\sum_{r=1}^{L_z} \exp\left(\langle \mathbf{b}_i^z, \mathbf{q}_r^z \rangle / \tau\right)} \tag{3}$$

where $\tau$ is the temperature parameter. It should be noted that patch prototypes should not only include new information, but also reserve history information. Therefore, the update gate $\psi$ Song et al. (2024) is employed to update $\mathbf{b}_i^z$ by

$$\mathbf{b}_i^z = \left(\mathbf{1}_D - \psi\right) \circ \mathbf{b}_i^z + \psi \circ \sum_{j=1}^{L_z} v_{ij}^z \mathbf{q}_j^z \tag{4}$$

where $\circ$ takes element-by-element multiplication,

$$\psi = \sigma\left(\mathbf{U}_1^z \mathbf{b}_i^z + \mathbf{U}_2^z \sum_{k=1}^{L_z} v_{ik}^z \mathbf{q}_k^z\right) \tag{5}$$

and $\sigma$ is the sigmoid activation function. $\mathbf{U}_1^z$ and $\mathbf{U}_2^z$ are learnable matrices which are initialized randomly to adjust the degree of preservation and removal of the history prototypes and new information.

**Query Reconstruction.** Within the scope of reconstruction, patch prototypes are expected to well reconstruct the query series. In each scale of $z$, the updated patch prototypes $\mathbf{B}^z$ are taken to reconstruct the query series $\mathbf{Q}^z$ to be $\hat{\mathbf{Q}}^z = \left\{\hat{\mathbf{q}}_1^z, \hat{\mathbf{q}}_2^z, \cdots, \hat{\mathbf{q}}_{L_z}^z\right\}$ according to

$$\hat{\mathbf{q}}_j^z = \sum_{k=1}^{M} w_{jk}^z \mathbf{b}_k, \quad \text{where} \quad w_{jk}^z = \frac{\exp\left(\langle \mathbf{q}_j^z, \mathbf{b}_k^z \rangle / \tau\right)}{\sum_{r=1}^{M} \exp\left(\langle \mathbf{q}_j^z, \mathbf{b}_r^z \rangle / \tau\right)} \tag{6}$$

is the attention weight of the query $\mathbf{q}_j^z$ for the patch prototype $\mathbf{b}_k^z$, $\mathbf{W}^z = \left( w_{jk}^z \right)_{L_z \times M}$.

Different scales of reconstructed queries $\hat{\mathbf{Q}}^{z_1}, \hat{\mathbf{Q}}^{z_2}, \cdots, \hat{\mathbf{Q}}^{z_m}$ should be integrated properly to reconstruct $\mathbf{Q}^z$ in consideration of different length of temporal dependencies. Recalling the strategy of average pooling, local information of $\mathbf{q}_t$ is included into $\mathbf{q}_{t_j}^{z_j}$ where $(t_j - 1) \cdot z_j + 1 \leq t \leq t_j \cdot z_j (j = 1, 2, \cdots, m)$. In other words, in the scale of $z_j$, $\mathbf{q}_{t_j}^{z_j}$ contains the normal information of $\mathbf{q}_t$. Naturally, $\hat{\mathbf{q}}_{t_j}^{z_j} (j = 1, 2, \cdots, m)$ is taken into account to reconstruct $\mathbf{q}_t$ (as illustrated in Figure 3(b)). All related $\hat{\mathbf{q}}_{t_j}^{z_j}$ is integrated with network to reconstruct $\mathbf{q}_t$:

$$\hat{\mathbf{q}}_t = Linear \left( ReLU \left( Linear \left( \hat{\mathbf{q}}_{t_i}^{z_1}, \cdots, \hat{\mathbf{q}}_{t_i}^{z_2}, \cdots, \hat{\mathbf{q}}_{t_i}^{z_m} \right) \right) \right). \tag{7}$$

Finally, $\hat{\mathbf{Q}} = (\hat{\mathbf{q}}_1, \hat{\mathbf{q}}_2, \cdots, \hat{\mathbf{q}}_L)$ is concatenated with $\mathbf{Q}^{z_1}$ and decoded into original space $\mathbb{R}^C$ as the reconstructed series $\hat{\mathbf{X}}^z$ of $\mathbf{X}$ by normal patch prototypes.

## 3.2 Learning Period Prototypes

Most time series in the real world are multi-periodic, and these periods influence each other, presenting the overall variation tendency of the time series Wu et al. (2023). In addition to normal patch prototypes, period prototypes of the normal time series are learned to model characteristics of the different periodic patterns (the right branch of the H-PAD framework in Figure 2).

**Period Division.** Technically, period division of the time series relies on the frequency information of the series in frequency domain. To this end, the time series $\mathbf{X} \in \mathbb{R}^{L \times C}$ is transformed into the frequency domain by Fast Fourier Transform (FFT) Wu et al. (2023) to get the averaged amplitude values by

$$\mathbf{A} = Avg \left( Amp \left( FFT(\mathbf{X}) \right) \right).$$

Considering the sparsity of frequency domain, only the top-$k$ amplitudes are selected to get $\mathbf{P} = \{p_1, p_2, \cdots, p_K\}$ for period partition where

$$\{f_1, f_2, \ldots f_K\} = argTopK(\mathbf{A}), \quad p_i = \left\lceil \frac{L}{f_i} \right\rceil (i = 1, 2, \cdots, K)$$

and $\{f_1, f_2, \ldots f_K\}$ is the most significant frequencies Zhou et al. (2022). With $p \in \mathbf{P}$, the time series $\mathbf{X}$ can be divided get $N = \left\lceil \frac{L}{p} \right\rceil$ segments (zero-padding at the end). As each observation variable has its own special changing period, the period prototype is expected to learn along each variable. Therefore, one univariate time series $\mathbf{X}_j \in \mathbb{R}^L$ taken from $\mathbf{X} \in \mathbb{R}^{L \times C}$ is divided into $N$ segments of length $p$ and is reshaped to $\mathbf{X}_j^p \in \mathbb{R}^{N \times p}$. Then, the time series $\mathbf{X} \in \mathbb{R}^{L \times C}$ is reorganized to be $\mathcal{X}^p = \{\mathbf{X}_1^p, \mathbf{X}_2^p, \cdots, \mathbf{X}_C^p\} \in \mathbb{R}^{N \times p \times C}$ for learning period prototypes.

**Update Period Prototypes.** Through Transformer Encoder, $\mathcal{X}^p$ is embedded to obtain $\mathbf{Q}^p$ (see Appendix B). For one observing variable of the time series, one period prototype $\mathbf{b}^p \in \mathbb{R}^D$ is learned from one period partition $\mathbf{Q}^p = \{\mathbf{q}_1^p, \mathbf{q}_2^p, \cdots, \mathbf{q}_N^p\}$ where $\mathbf{Q}^p \in \mathcal{Q}^p$ (shown as in Figure 3(c)). With randomly initialized $\mathbf{b}^p$, the period prototype is updated with weighted segmented periods by

$$\mathbf{b}^p = (\mathbf{1}_D - \psi) \circ \mathbf{b}^p + \psi \circ \sum_{j=1}^{N} v_j^p \mathbf{q}_j^p \tag{8}$$

where

$$\psi = \sigma \left( \mathbf{U}_1^p \mathbf{b}^p + \mathbf{U}_2^p \sum_{k=1}^{N} v_k^p \mathbf{q}_k^p \right), \qquad v_j^p = \frac{\exp \left( \langle \mathbf{b}^p, \mathbf{q}_j^p \rangle / \tau \right)}{\sum_{r=1}^{N} \exp \left( \langle \mathbf{b}^p, \mathbf{q}_r^p \rangle / \tau \right)} \tag{9}$$

where $\mathbf{U}_1^p$ and $\mathbf{U}_2^p$ are also learnable matrices just like in Eq.5. Finally, a set of period prototypes for $C$ variables are gained, $\mathbf{B}^p = \{\mathbf{b}_1^p, \mathbf{b}_2^p, \cdots, \mathbf{b}_C^p\}$ for query reconstruction.

**Query Reconstruction.** Considering the variable correlations, each query vector $\mathbf{q}_i^p \in \mathbf{Q}^p$ can be recovered to be $\hat{\mathbf{q}}_i^p$ with the updated period prototypes $\mathbf{B}^p$ according to

$$\hat{\mathbf{q}}_i^p = \sum_{j=1}^{C} w_{ij}^p \mathbf{b}_j^p, \qquad \text{where} \quad w_{ij}^p = \frac{\exp\left(\langle \mathbf{q}_i^p, \mathbf{b}_j^p \rangle / \tau\right)}{\sum_{n=1}^{C} \exp\left(\langle \mathbf{q}_i^p, \mathbf{b}_n^p \rangle / \tau\right)}. \tag{10}$$

The reconstructed period partition $\hat{\mathbf{Q}}^p = \{\hat{\mathbf{q}}_1^p, \hat{\mathbf{q}}_2^p, \cdots, \hat{\mathbf{q}}_N^p\}$ are concatenated with $\mathbf{Q}^p = \{\mathbf{q}_1^p, \mathbf{q}_2^p, \cdots, \mathbf{q}_N^p\}$ to produce $\tilde{\mathbf{Q}}^p = \{\tilde{\mathbf{q}}_1^p, \tilde{\mathbf{q}}_2^p, \cdots, \tilde{\mathbf{q}}_N^p\}$, where $\tilde{\mathbf{q}}_i^p = (\mathbf{q}_i^p, \hat{\mathbf{q}}_i^p) \in \mathbb{R}^{2D}$. Collecting all reconstructed partitions of $C$ variables, $\tilde{\mathcal{Q}}^p = \left\{\tilde{\mathbf{Q}}_1^p, \tilde{\mathbf{Q}}_2^p, \cdots, \tilde{\mathbf{Q}}_C^p\right\}$ is used to reconstruct $\mathbf{X}$ as $\hat{\mathbf{X}}_i^p$ with decoder, where $\hat{\mathbf{X}}_i^p$ is the reconstructed data obtained in the $i$-th period. The reconstructed data of the final period branch is $\hat{\mathbf{X}}_p = \sum_{i=1}^{K} \hat{\mathbf{X}}_i^p$.

After finishing reconstruction with patch and period prototypes respectively, $\mathbf{X}$ is finally reconstructed by the weighted summation of $\hat{\mathbf{X}}_z$ and $\hat{\mathbf{X}}_p$, $\hat{\mathbf{X}} = \gamma \hat{\mathbf{X}}_z + (1 - \gamma) \hat{\mathbf{X}}_p$, where $\gamma$ is a hyperparameter.

### 3.3 Loss Function and Anomaly Scores

Generally, the reconstruction loss is surely one component of the loss function for training phase, formulated as

$$L_{rec} = \|\mathbf{X} - \hat{\mathbf{X}}\|_F \tag{11}$$

where $\|\cdot\|_F$ is the Frobenius norm. Besides, too many prototypes for reconstruction may over-interpret normal information in the training time series. To make sure that the most related prototypes are presented in reconstructing, the sparsity constraints on reconstruction weights are required to reduce the likelihood of over-generalization problem. In this paper, the entropy loss is taken as the sparsity constraint on reconstruction weights Song et al. (2024):

$$L_{ent} = \sum_{z=z_1}^{z_m} \sum_{j=1}^{L_z} \sum_{i=1}^{M} -w_{ji}^z \log\left(w_{ji}^z\right). \tag{12}$$

Note that the sparsity is not required for period reconstruction since all period prototypes are needed for reconstruction. Specially for period prototypes, they must appropriately characterize the periodicities of the training time series as much as possible. Therefore, the period loss in feature space is designed based on the distance between the period prototype $\mathbf{b}_i^p$ and the query vector $\mathbf{q}_{ij}^p$:

$$L_{prd} = \sum_{m=1}^{K} \sum_{i=1}^{C} \sum_{j=1}^{N} \|\mathbf{b}_i^{p_m} - \mathbf{q}_{ij}^{p_m}\|_2 \tag{13}$$

where $\mathbf{q}_{ij}^p$ is the $j$-th query of the $i$-th variable with period $p$ in feature space $\mathbb{R}^D$. To sum up, the total loss function is a weighted combination of Eq.(11), Eq.(13) and Eq.(12):

$$LOSS = \alpha_1 L_{rec} + \alpha_2 L_{ent} + \alpha_3 L_{prd} \tag{14}$$

where $\alpha_1$, $\alpha_2$, and $\alpha_3$ are adaptive parameters of different loss parts.

To detecting anomalies, the protuberant deviations in both input space and feature space are designed into anomalies scores. In the input space, the reconstruction error is generally considered to be anomalies scores:

$$s_r(t) = \|\hat{\mathbf{x}}_t - \mathbf{x}_t\|_2. \tag{15}$$

In the feature space, for patch reconstruction of $\mathbf{x}_t$, the distance between the corresponding query $\mathbf{q}_t$ and the query's nearest prototype $\mathbf{b}_{sim}^{z_j}$ in the scale $z_j$ is considered to be included into the anomaly score of $\mathbf{x}_t$:

$$s_z(t) = \sum_{j=1}^{m} \frac{1}{z_j} \left\| \mathbf{q}_t - \mathbf{b}_{sim}^{z_j} \right\|_2 \tag{16}$$

where $\mathbf{q}_t = Encoder(\mathbf{x}_t)$. The larger the scale $z_j$, the more different the patch prototype of scale $z_j$ from the original query. An inverse proportional parameter $\frac{1}{z_j}$ is employed to adapt the influence of different scale of the patch prototypes on the score $s_z(t)$. For period reconstruction of $\mathbf{x}_t$, the anomaly score of $\mathbf{x}_t$ is related with the period where it locates:

$$s_p(t) = \sum_{k=1}^{K} \sum_{i=1}^{C} \left\| Encoder \left( O_{i,\rho(t)}^{p_k} \right) - \mathbf{b}_i^{p_k} \right\|_2 \tag{17}$$

where $\rho(t) = \left\lceil \frac{t}{p_k} \right\rceil$, and $O_{i,\rho(t)}^{p_k} \in \mathbb{R}^{p_k}$ is the $\rho(t)$-th period of length $p_k$ along the $i$-th variable. $\mathbf{b}_i^{p_k}$ is the period prototype of length $p_k$ in $i$-th variable. Thus the distance of the $\rho(t)$-th encoded period with the learned period prototype 17.

Considering that both scores in feature space affects the reconstruction error, they are integrated into

$$s(t) = softmax \left( s_z(t) + \beta s_p(t) \right) \times s_r(t) \tag{18}$$

where $\beta$ is the weight adapting parameters.

## 4 EXPERIMENTS

### 4.1 EXPERIMENTAL SETUP

Our model H-PAD is evaluated on seven real-world multivariate datasets, namely, **MSL**, **SMAP**, **PSM**, **SMD**, **SWaT**, **NIPS_TS_Water** and **NIPS_TS_Swan**. The more detailed content of the dataset can be found in Appendix C. In addition, specific implementation details are given in Appendix D.

### 4.2 MAIN RESULTS

We comprehensively compare our model with 16 baseline models. Table 1 gives the evaluation results of different baseline models and our model in five real datasets. It can be seen that our model H-PAD can achieve the best results in most datasets, with an F1 score of more than 95% on all datasets.

However, many works have demonstrated that PA can lead to faulty performance evaluations (Wang et al., 2023; Kim et al., 2021; Huet et al., 2022), and it is known that using PA can result in state-of-the-art performance even with random scores or random initialized non-trained models, making it impossible to conduct a fair comparison and assess the effectiveness of the models. To ensure a fair comparison between H-PAD and the baseline models, we used AUC-ROC and AUC-PR as evaluation metrics. As shown in Table 2, comparison of H-PAD with other reconstruction models, H-PAD achieves the best or second-best results on most datasets. Furthermore, H-PAD exhibited the highest average AUC-ROC score and the highest AUC-PR score in all seven datasets, highlighting its effectiveness. Please refer to Appendix E for a more detailed description of the evaluation criteria.

### 4.3 ABLATION STUDY

**Effectiveness of module** The different effectiveness of using patch prototypes and period prototypes is studies. As shown in Table 3, no matter which prototype is removed, the performance will decrease, and the

Table 1: Precision, recall, F1-score results (as %) on five real-world datasets. 'A.T.' means Anomaly Transformer. 'Avg' means average. The best results are marked in bold, and the second best results are marked in underline.

| Model | MSL | | | SMAP | | | PSM | | | SMD | | | SWaT | | | Avg |
|---|---|---|---|---|---|---|---|---|---|---|---|---|---|---|---|---|
| | Pre | Rec | F1 | Pre | Rec | F1 | Pre | Rec | F1 | Pre | Rec | F1 | Pre | Rec | F1 | F1 |
| Isolation Forest | 53.94 | 86.54 | 66.45 | 52.39 | 59.07 | 55.53 | 76.09 | 92.45 | 83.48 | 42.31 | 73.29 | 53.64 | 49.29 | 44.95 | 47.02 | 61.22 |
| OC-SVM | 59.78 | 86.87 | 70.82 | 53.85 | 59.07 | 56.34 | 62.75 | 80.89 | 70.67 | 44.34 | 76.72 | 56.19 | 45.39 | 49.22 | 47.23 | 60.25 |
| LOF | 47.72 | 85.25 | 61.18 | 58.93 | 56.33 | 57.60 | 57.89 | 90.49 | 70.61 | 56.34 | 39.86 | 46.68 | 72.15 | 65.43 | 68.62 | 60.94 |
| DAGMM | 89.60 | 63.93 | 74.62 | 86.45 | 56.73 | 68.51 | 93.49 | 70.03 | 80.08 | 67.30 | 49.89 | 57.30 | 89.92 | 57.84 | 70.40 | 70.18 |
| MMPCACD | 81.42 | 61.31 | 69.95 | 88.61 | 75.84 | 81.73 | 76.25 | 78.35 | 77.29 | 71.20 | 79.28 | 75.02 | 82.52 | 68.29 | 74.73 | 75.74 |
| Deep-SVDD | 91.92 | 76.63 | 83.58 | 89.93 | 56.02 | 69.04 | 95.41 | 86.49 | 90.73 | 78.54 | 79.67 | 79.10 | 80.42 | 84.45 | 82.39 | 80.97 |
| THOC | 88.45 | 90.97 | 89.69 | 92.06 | 89.34 | 90.68 | 88.14 | 90.99 | 89.54 | 79.76 | 90.95 | 84.99 | 83.94 | 86.36 | 85.13 | 88.01 |
| ITAD | 69.44 | 84.09 | 76.07 | 82.42 | 66.89 | 73.85 | 72.80 | 64.02 | 68.13 | 86.22 | 73.71 | 79.48 | 63.13 | 52.08 | 57.08 | 70.92 |
| LSTM-VAE | 85.49 | 79.94 | 82.62 | 92.20 | 67.75 | 78.10 | 73.62 | 89.92 | 80.96 | 75.76 | 90.08 | 82.30 | 76.00 | 89.50 | 82.20 | 81.24 |
| OmniAnomaly | 89.02 | 86.37 | 87.67 | 92.49 | 81.99 | 86.92 | 88.39 | 74.46 | 80.83 | 83.68 | 86.82 | 85.22 | 81.42 | 84.30 | 82.83 | 84.69 |
| InterFusion | 81.28 | 92.70 | 86.62 | 89.77 | 88.52 | 89.14 | 83.61 | 83.45 | 83.52 | 87.02 | 85.43 | 86.22 | 80.59 | 85.58 | 83.01 | 85.70 |
| BeatGAN | 89.75 | 85.42 | 87.53 | 92.38 | 55.85 | 69.61 | 90.30 | 93.84 | 92.04 | 72.90 | 84.09 | 78.10 | 64.01 | 87.46 | 73.92 | 80.24 |
| A.T. | 91.88 | 92.98 | 92.43 | 93.65 | 99.47 | 96.47 | 95.86 | 98.77 | 97.29 | 89.45 | 94.36 | 91.84 | 90.98 | 95.56 | 92.41 | 94.09 |
| DCdetector | 92.09 | 98.89 | 95.37 | 94.42 | 98.95 | 96.63 | 97.24 | 97.72 | 97.48 | 86.08 | 85.60 | 85.84 | 93.29 | 100.00 | 96.53 | 94.37 |
| D3R | 91.77 | 94.33 | 93.03 | 92.23 | 96.11 | 94.21 | 93.84 | 99.11 | 96.45 | 87.74 | 96.09 | 91.91 | 83.09 | 83.00 | 83.04 | 91.73 |
| MEMTO | 91.95 | 97.23 | 94.56 | 93.66 | 99.73 | 96.60 | 97.47 | 98.60 | 98.03 | 88.24 | 96.16 | 92.03 | 94.28 | 91.72 | 93.73 | 94.99 |
| DMamba | 93.69 | 64.06 | 76.09 | 95.10 | 52.98 | 68.05 | 98.66 | 82.59 | 89.91 | 92.57 | 54.04 | 68.24 | 94.11 | 86.75 | 90.28 | 78.51 |
| GSC_MAD | 94.19 | 93.09 | 93.63 | 89.57 | 98.35 | 93.76 | 97.97 | 99.14 | 98.89 | 92.25 | 94.42 | 93.32 | 96.73 | 95.11 | 95.91 | 95.10 |
| **H-PAD** | 94.05 | 96.88 | **95.45** | **96.00** | 98.45 | **97.21** | **98.82** | **99.41** | **99.12** | 92.86 | 98.20 | 95.45 | 94.34 | 100.00 | **97.09** | **96.86** |

Table 2: AUC-ROC and AUC-PR on seven real-world datasets. 'N_T_W' means NIPS_TS_Water. 'N_T_S' means NIPS_TS_Swan. 'AR' means AUC-ROC. 'AP' means AUC-PR.

| Model | MSL | | SMAP | | PSM | | SMD | | SWaT | | N_T_W | | N_T_S | | Avg | |
|---|---|---|---|---|---|---|---|---|---|---|---|---|---|---|---|---|
| | AR | AP | AR | AP | AR | AP | AR | AP | AR | AP | AR | AP | AR | AP | AR | AP |
| LSTM-VAE | 52.12 | 4.52 | 50.83 | 4.19 | 49.15 | 40.22 | 50.05 | 4.15 | 49.59 | 4.13 | 51.75 | 4.34 | 51.73 | 4.49 | 50.74 | 9.43 |
| D3R | **65.26** | **16.99** | 41.35 | 10.62 | 50.03 | 26.31 | 64.20 | 12.24 | 56.65 | 13.30 | 80.32 | 12.39 | 53.40 | 40.97 | 58.74 | 18.97 |
| A.T. | 48.72 | 10.64 | 49.67 | 12.50 | 48.56 | 29.42 | 47.28 | 3.70 | 29.40 | 8.82 | 33.46 | 1.48 | 43.49 | 28.62 | 42.94 | 13.59 |
| DCdetector | 50.06 | 10.61 | 48.87 | 12.48 | 49.83 | 27.64 | 48.77 | **41.16** | 49.74 | 11.60 | 50.53 | 1.72 | 48.50 | 31.71 | 49.47 | 19.56 |
| MEMTO | 49.99 | 10.48 | **59.59** | **16.29** | 49.75 | 26.96 | 73.24 | 10.35 | 45.41 | 11.45 | 60.96 | 4.21 | 51.12 | 49.06 | 55.72 | 18.40 |
| DMamba | 61.54 | 15.02 | 38.99 | 10.85 | 59.53 | 40.17 | 64.55 | 11.99 | 74.49 | 25.33 | 96.93 | 46.32 | 77.84 | 64.64 | 67.69 | 30.61 |
| **H-PAD** | 59.99 | 15.06 | 59.13 | 15.30 | **75.01** | **51.83** | 76.49 | 14.05 | 81.54 | 53.99 | 75.96 | 7.30 | **81.66** | **74.31** | **72.83** | **33.12** |

performance decrease is the largest when the patch prototype is removed. Based on the comparison of these results, the effectiveness of using patch prototypes and period prototypes is demonstrated.

**Abnormality Criteria** Effectiveness of different abnormality score criteria is also studied. As shown in Table 4, when we only used the reconstruction error as the criterion, the F1 score dropped the most, with an average drop of 14.21%. Removing different evaluation criteria separately will result in different degrees of decline in results, which also proves the effectiveness of the abnormality criteria we used in this paper.

## 4.4 DISCUSSION AND ANALYSIS

**Parameter sensitivity analysis** The impact of the numbers of prototypes, scales, and periods on H-PAD is analyzed numerically. It can be seen from Figure 4 that F1 has small variance for different number of prototypes per scale (Figure 4(a)) to exhibit the robustness of H-PAD, as well as the number of periods shown in Figure 4(c) . H-PAD can get optimal F1 for each dataset (Figure 4(b)) with the best scale number.

Table 3: Modules Effectiveness measured in F1-score (%). **Scale** is the patch prototype branch, and **Period** is the period prototype branch.

| Scale | Period | MSL | SMAP | PSM | SWaT | SMD | Avg. |
|:---:|:---:|---|---|---|---|---|---|
| ✗ | ✗ | 93.93 | 96.30 | 97.95 | 93.73 | 92.03 | 94.78 |
| ✓ | ✗ | 94.56 | 96.51 | 98.37 | 96.89 | 94.26 | 96.11 |
| ✗ | ✓ | 83.48 | 69.52 | 93.72 | 89.55 | 78.13 | 83.08 |
| ✓ | ✓ | **95.45** | **97.21** | **99.12** | **97.09** | **95.45** | **96.86** |

Table 4: Effectiveness of anomaly criterions. Comparison using F1 score (%).

| $s_{rec}$ | $s_z$ | $s_p$ | MSL | SMAP | PSM | SWaT | SMD | Avg. |
|:---:|:---:|:---:|---|---|---|---|---|---|
| ✓ | ✗ | ✗ | 88.32 | 78.21 | 93.40 | 94.08 | 59.25 | 82.65 |
| ✓ | ✓ | ✗ | 93.18 | 96.73 | 98.24 | 96.78 | 93.24 | 95.63 |
| ✓ | ✗ | ✓ | 91.27 | 91.71 | 93.40 | 84.88 | 59.61 | 84.17 |
| ✗ | ✓ | ✓ | 93.18 | 96.47 | 97.85 | 96.62 | 89.84 | 94.79 |
| ✓ | ✓ | ✓ | **95.45** | **97.21** | **99.12** | **97.09** | **95.45** | **96.86** |

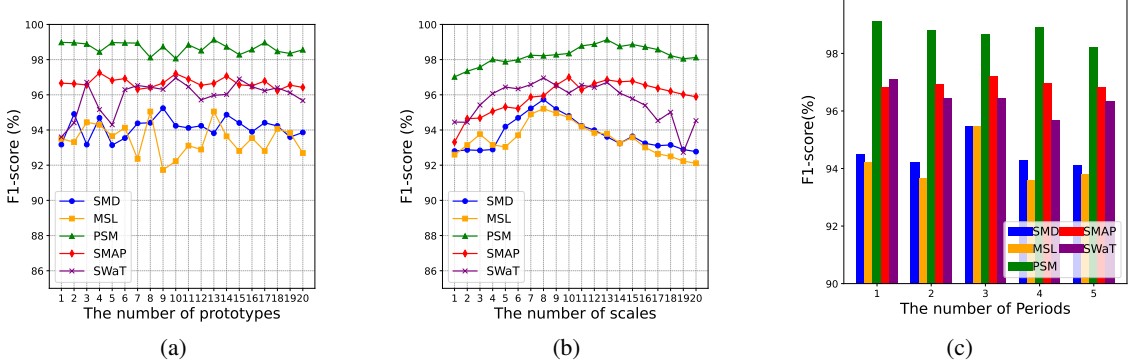

(a)  (b)  (c)

Figure 4: Sensitivity analysis of hyperparameters for H-PAD.

## 5 CONCLUSION

This paper introduces a new time series anomaly detection model called H-PAD. The model utilizes different patch prototypes and period prototypes to detect various interval and period anomalies. Furthermore, we employ data of different scales to capture short-term changes and use data of different periods to capture periodic information, enabling the modeling of the time series using long-term and short-term normal prototypes. H-PAD is compared on seven datasets, illustrating the advantages of the model. In future work, we plan to optimize the overall framework to improve efficiency, reducing training time and memory consumption without compromising performance.

ACKNOWLEDGMENTS

This work is supported by the National Natural Science Foundation of China (61672205), the High-Level Talents Research Start-Up Project of Hebei University (521100222002), the Innovation Capacity Enhancement Program Science and Technology Platform Project of Hebei Province (22567623H) and the Natural Science Foundation of Hebei Province (No. F2024201032, A2024201031 and H2024201062).

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

## A PROBLEM DESCRIPTION

Figure 5(a) visualizes the reconstructed series by Anomaly Transformer Xu et al. (2022). It can be seen that both normal and abnormal points are well reconstructed by trained Anomaly Transformer, resulting the over-generalization problem that anomalies can no longer be identified by high reconstruction errors. To fight against over-generalization, MEMTO employs a memory bank of the normal point prototypes to help reconstruct time series Song et al. (2024). The point prototypes are learned from normal series to depict the normal points characters. In inference phrase, the learned point prototypes are taken to reconstruct the test series. Because anomalies are absent from the prototype learning, MEMTO reconstructs the test series with none abnormal information, thereby alleviating the over-generalization problem. However, as visualized in Figure 5(b), MEMTO does not properly solve the over-generalization problem. The main reason lies in that point prototypes can not exhibit the intrinsic relevancy of neighbored points. Normal points and abnormal points are treated equally for reconstruction, indicating that the learned point prototypes can not well distinguish the abnormal characteristics from normal characteristics. What's more, point prototypes can not exhibit the varying trend in an interval, failing to identify interval anomalies which manifest as brief data fluctuations or sudden changes over a short interval. As a long-term anomalies, period anomalies can also not be identified by point prototypes since single-point normal prototypes typically cannot capture periodic regularities. After reconstruction using different normal prototypes in H-PAD, the reconstructed data is closer to the normal data and farther from the anomalous data (as shown in Figure 6).

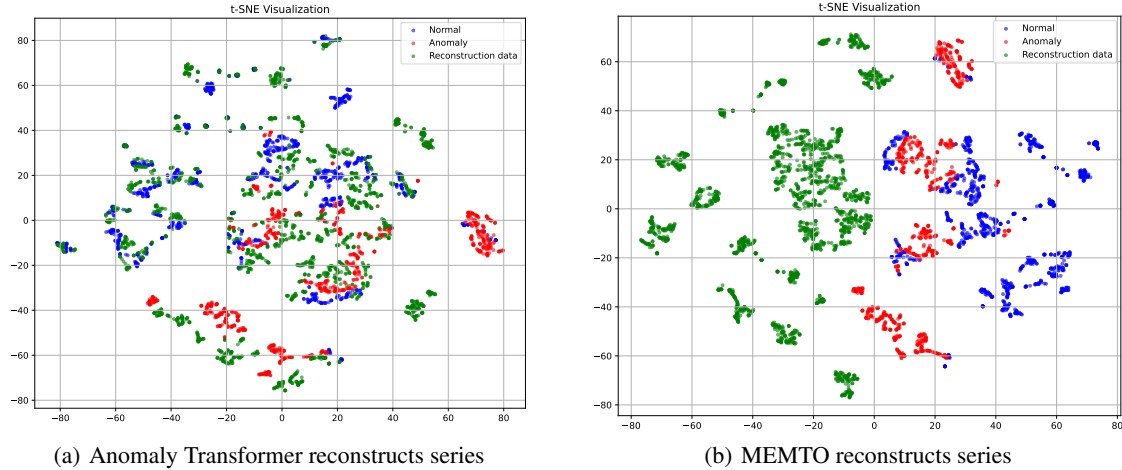

(a) Anomaly Transformer reconstructs series          (b) MEMTO reconstructs series

Figure 5: Figure a is t-SNE graph of the original data and the data reconstructed by Anomaly Transformer, and Figure b is t-SNE graph of the original data and the data reconstructed by MEMTO. Blue points are normal data points, red points are abnormal data points, and green points are reconstructed data points.

## B ENCODER

To make the data operation clearer, this section explains the working principle of the encoder part.

To learn prototypes in feature space, different scale series are embedded into a high-dimensional feature space with Transformer Encoder (Figure 7):

$$\mathbf{Q}^z = Transformer\left(Embedding\left(\mathbf{X}^z\right)\right) \tag{19}$$

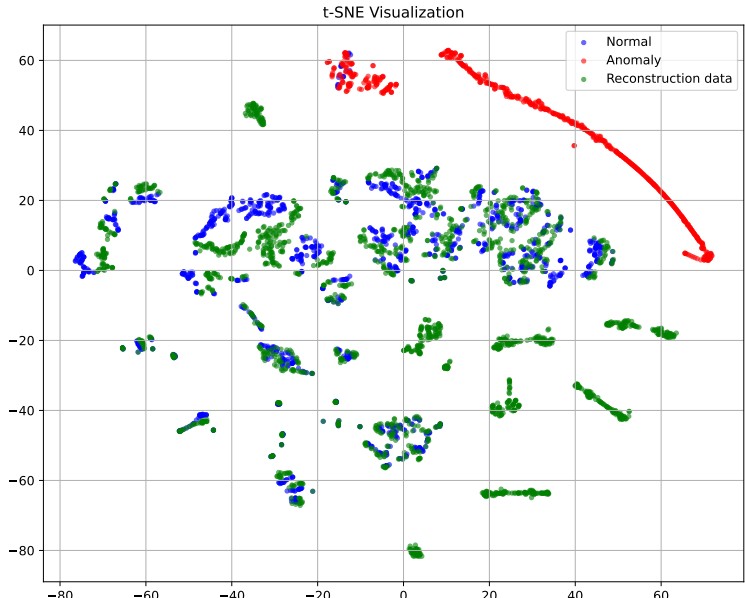

Figure 6: The t-SNE graphs of the original data and the data reconstructed by H-PAD.

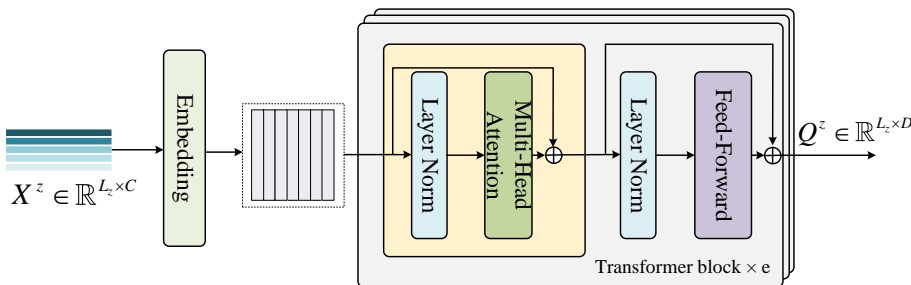

Figure 7: Learn temporal dependencies at different scales with multiple Transformers.

where $\mathbf{Q}^z = \left\{\mathbf{q}_1^z, \mathbf{q}_2^z, \cdots, \mathbf{q}_{L_z}^z\right\} \in \mathbb{R}^{L_z \times D}\,(D > C)$. Specifically, the pooled series $\mathbf{X}^z$ is embedded into a high-dimensional space through a linear layer in order to present more complex information and latent regularities. Followed with multiple layers of Transformer blocks which capture long-term temporal dependencies, the query series $\mathbf{Q}^z$ is produced for learning the patches prototypes in different scales.

To characterize each period, $\mathbf{X}_j^p \in \mathbb{R}^{N \times p}$ is embedded into a high-dimensional features space $\mathbb{R}^{N \times D}$ by a linear network and then Transformer blocks learn the variable correlations according to (Figure 8)

$$\mathcal{Q}^p = Transformer\left(Embedding\left(\mathbf{X}_1^p, \mathbf{X}_2^p, \cdots, \mathbf{X}_C^p\right)\right) \tag{20}$$

where $\mathcal{Q}^p = \{\mathbf{Q}_1^p, \mathbf{Q}_2^p, \cdots, \mathbf{Q}_C^p\}$ and $\mathbf{Q}_j^p \in \mathbb{R}^{N \times D}$. $\mathcal{Q}^p$ is submitted to the period query update module to learn one period prototypes for each variable. Note that one encoder is structured for one period division learning since different periods $p$ of the time series present different variable correlations.

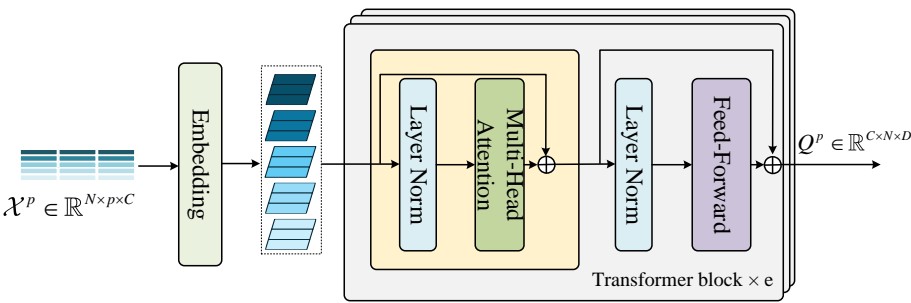

Figure 8: Transformer Encoder for learning multiple periods prototypes.

## C DATASETS

Our model H-PAD is evaluated on seven real-world multivariate datasets. The specific description of the dataset is shown in Table 5.

Table 5: The dataset used in this study. Train and Test represent the number of time points in the training dataset and the test dataset, respectively. AR represents the anomaly rate.

| Dataset | Dims | Train | Test | AR (%) |
|---|---|---|---|---|
| SWaT | 51 | 475200 | 449919 | 12.14 |
| PSM | 25 | 132418 | 87841 | 27.76 |
| MSL | 55 | 58317 | 73729 | 10.48 |
| SMAP | 25 | 135183 | 427617 | 12.83 |
| SMD | 38 | 708405 | 708420 | 4.16 |
| NIPS_TS_Water | 9 | 69260 | 69261 | 1.1 |
| NIPS_TS_Swan | 38 | 60000 | 60000 | 32.60 |

- **SWaT**(Secure Water Treatment) Li et al. (2019) is a collection of sensor data from a water treatment process running continuously for 11 days under various operating conditions.

- **PSM**(Pooled Server Metrics) Abdulaal et al. (2021) is a public data set composed of data generated by different application servers in eBay.

- **MSL**(Mars Science Laboratory rover) Hundman et al. (2018) is collected by NASA shows the status data of sensor and actuator of the Mars rover.

- **SMAP**(Soil Moisture Active Passive satellite) Hundman et al. (2018) is provided by NASA's Soil Moisture Active Passive Satellite mission, which collects soil moisture data on a global scale.

- **SMD**(Server Machine Dataset) Su et al. (2019) is server data collected by a large Internet company over a period of 5 weeks and contains telemetry data from multiple servers.

- **NIPS_TS_Water** Lai et al. (2021) originates from a water quality monitoring system, which records water quality indicators measured by multiple sensors.

- **NIPS_TS_Swan** Rehbach et al. (2018) is a multivariate time series extracted from vector magnetograms of the solar photosphere.

## D IMPLEMENTATION DETAILS

In this paper, H-PAD algorithm experiments, parameters sensitivity analysis and ablation studies are implemented in PyTorch using a single NVIDIA GeForce RTX 4090 24GB GPU. H-PAD uses the Adam optimizer to optimize the networks' parameters, with an initial learning rate of $10^{-4}$. The H-PAD training process uses early stopping mechanism that the model training stops when the model loss ($LOSS$) in Eq.(14) does not decrease within 8 epochs. We use a non-overlapping sliding window of length 100 to split the time series of each dataset and divide the training dataset into 80% training set and 20% validation set. The batch size is set to 32, and the model processes 32 batches of data each time. To find the best results, we set different hyperparameters for different datasets, such as the feature dimension $D$, the number $z$ of scales, the number $K$ of periods, and the number $M$ of patch prototypes. First, the hyperparameters $(D, z, K, M)$ are set to $(128, 5, 3, 20)$, and then the optimal parameters for each dataset are obtained by adjusting the parameters. For the baseline model for comparison, we reproduced it using the parameters given in its paper. Our model defines anomalies as time points where the anomaly score exceeds a hyperparameter threshold $\delta$, and marks the label of the anomaly point as 1.

## E EVALUATION CRITERIA

Considering the main task of MTSAD, H-PAD is trained with only normal series and expected to be able to detect anomalies in the testing series, which is also considered to be an unbalanced binary classification problem. Thus, to assess and compare the performance of the proposed H-PAD, evaluation criteria based on confusion matrix are used in this study. Confusion matrix is composed of True Positive (TP), True Negative (TN), False Positive (FP) and False Negative (FN). Note that abnormal observations are regarded as positive samples in MTSAD. Based on the confusion matrix, several evaluation metrics can be employed to quantify the detection performance:

- **Precision** is the proportion of observations predicted correctly to be abnormal among those predicted to be abnormal, calculated as:

$$\text{Pre} = \frac{TP}{TP + FP}. \tag{21}$$

- **Recall** is the proportion of observations predicted correctly to be abnormal among those labeled to be abnormal, calculated as:

$$\text{Rec} = \frac{TP}{TP + FN}. \tag{22}$$

- **F1-score** can comprehensively evaluate the detecting performance. It is the harmonic mean of precision and recall, calculated as:

$$\text{F1} = 2 \cdot \frac{\text{Pre} \cdot \text{Rec}}{\text{Pre} + \text{Rec}}. \tag{23}$$

- **AUC-ROC** (Area Under the Receiver Operating Characteristic Curve) is the area under the ROC (Receiver Operating Characteristic) curve which depicts the relationship between the false positive rate (FPR) and the true positive rate (TPR) where

$$TPR = \frac{TP}{TP + FN}, \quad and \quad FPR = \frac{FP}{FP + TN} \tag{24}$$

- **AUC-PR** (Area Under the Precision-Recall Curve) is the area under the Precision-Recall curve which is more suitable to evaluate imbalanced classifiers.

## F  TRAINING EFFICIENCY ANALYSIS

The efficiency of the H-PAD training model is compared with another memory model MEDMTO. The results are shown in Table 6. Since H-PAD learns patch prototypes of time series of different scales and period prototypes of different periods, its efficiency is much higher than MEMTO. MACs is the total number of multiplication-accumulation operations, which is used to measure the computational complexity of the neural network model. Epoch Time is the training time per epoch, in seconds. Max Memory Allocated is the maximum GPU memory usage during training. Total Parameters is the total number of parameters in the neural network model, including weights, biases, and other parameters.

Table 6: Training efficiency analysis.

| Dataset | Method | MACs | NPARAMS | EROCH TIME | MAX MEMORY(GB) |
|---|---|---|---|---|---|
| MSL | MEMTO | 10415226880 | 5955182 | 1.46 | 1.04 |
|  | H-PAD | 90085779336 | 36345527 | 10.91 | 10.98 |
| SMAP | MEMTO | 10261626880 | 5862962 | 1.39 | 1.78 |
|  | H-PAD | 35983470152 | 23556227 | 19.58 | 9.36 |
| PSM | MEMTO | 10261626880 | 5862962 | 1.05 | 2.58 |
|  | H-PAD | 32452186440 | 22462860 | 10.70 | 3.35 |
| SWaT | MEMTO | 10394746880 | 5942886 | 3.65 | 4.37 |
|  | H-PAD | 72704662360 | 35411680 | 54.86 | 4.72 |
| SMD | MEMTO | 10328186880 | 5902924 | 11.17 | 1.47 |
|  | H-PAD | 41111111808 | 20404825 | 81.89 | 6.33 |

## G  MORE EXPERIMENTS

To further demonstrate the effectiveness of using patch prototypes and using period prototypes, we compared multiple indicators with the memory model MEMTO (Song et al., 2024) that only learns point normal prototypes. The results are shown in Table 7. Affiliation precision(Aff-P) and recall(Aff-R) are calculated based on the distance between ground truth and prediction events. VUS metric takes anomaly events into consideration based on the receiver operator characteristic(ROC) curve. R_A_R and R_A_P are Range-AUC-ROC and Range-AUC-PR, respectively, representing the two scores obtained under the ROC curve and PR curve according to the label transformation. V_ROC and V_PR are the volumes under the ROC curve and PR curve, respectively.

Table 7: Comparion results with MEMTO with different metrics on real-world datasets.

| Dataset | Method | Acc | F1 | Aff-P | Aff-R | R_A_R | R_A_P | V_ROC | V_PR |
|---|---|---|---|---|---|---|---|---|---|
| MSL | MEMTO | 98.09 | 93.54 | 51.43 | 96.00 | 90.56 | 88.35 | 88.71 | 86.73 |
|  | H-PAD | **99.03** | **95.45** | **55.98** | **96.25** | **91.34** | **88.66** | **89.94** | **87.91** |
| SMAP | MEMTO | 99.07 | 96.60 | **52.89** | 96.92 | 94.45 | 93.48 | 93.20 | 92.40 |
|  | H-PAD | **99.28** | **97.21** | 52.46 | **98.87** | **96.83** | **94.13** | **95.86** | **93.30** |
| PSM | MEMTO | 98.79 | 98.03 | 56.86 | 74.00 | 89.69 | **94.20** | 88.71 | **92.57** |
|  | H-PAD | **99.51** | **99.12** | **64.29** | **84.79** | **92.91** | 92.42 | **90.65** | 91.70 |
| SWaT | MEMTO | 98.44 | 93.73 | 59.04 | 93.41 | 92.01 | 89.16 | 92.09 | 89.23 |
|  | H-PAD | **99.22** | **97.09** | **60.40** | **97.54** | **98.28** | **95.88** | **98.31** | **95.91** |
| SMD | MEMTO | 99.18 | 92.03 | 56.19 | 86.93 | 74.69 | 71.21 | 74.95 | 71.48 |
|  | H-PAD | **99.60** | **95.45** | **68.91** | **93.22** | **81.02** | **78.86** | **82.02** | **79.85** |

In addition, to further explore the impact of different hyperparameters on model performance, we conducted more parameter sensitivity experiments. The results are shown in the table 8, table 9, table 10, table 11, table 12, figure 10(a), figure 10(b) and figure 9.

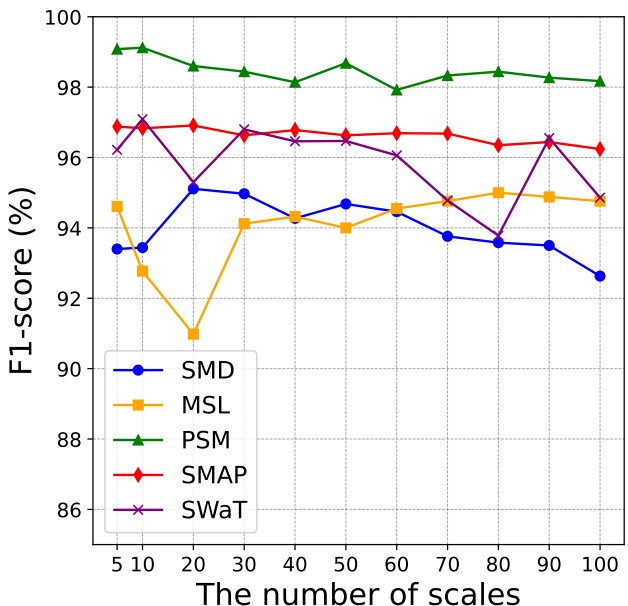

Figure 9: Sensitivity experiments with a larger number of prototypes.

Table 8: The impact of different sizes of hyperparameters $\gamma$ on the model. The result is F1 score (%).

| $\gamma$ | MSL | SMAP | PSM | SWaT | SMD |
|---|---|---|---|---|---|
| 0.1 | 94.06 | 97.01 | 98.99 | 96.33 | 89.37 |
| 0.3 | 93.14 | 96.93 | 98.95 | 94.43 | 93.80 |
| 0.5 | 95.45 | 97.21 | 99.12 | 97.09 | 95.45 |
| 0.7 | 93.20 | 96.30 | 98.97 | 96.68 | 94.16 |
| 0.9 | 92.69 | 96.28 | 99.04 | 95.97 | 95.15 |

Table 9: The impact of different sizes of hyperparameters $\beta$ on the model. The result is F1 score (%).

| $\beta$ | MSL | SMAP | PSM | SWaT | SMD |
|---|---|---|---|---|---|
| 1 | 93.07 | 95.33 | 98.23 | 95.01 | 91.33 |
| 0.1 | 93.99 | 96.72 | 98.77 | 96.39 | 93.47 |
| 0.01 | 95.45 | 97.21 | 99.12 | 97.09 | 95.45 |
| 0.001 | 94.93 | 97.01 | 99.05 | 96.87 | 94.69 |

Table 10: The impact of different sizes of hyperparameters $\alpha_1$ on the model. The result is F1 score (%).

| $\alpha_1$ | MSL | SMAP | PSM | SWaT | SMD |
|---|---|---|---|---|---|
| 0 | 94.92 | 96.83 | 98.13 | 96.46 | 90.95 |
| 0.1 | 93.32 | 96.59 | 98.13 | 94.66 | 95.00 |
| 0.5 | 94.00 | 96.63 | 98.68 | 94.66 | 93.59 |
| 0.8 | 93.53 | 96.42 | 98.95 | 96.48 | 94.61 |
| 1 | 95.45 | 97.21 | 99.12 | 97.09 | 95.45 |

Table 11: The impact of different sizes of hyperparameters $\alpha_2$ on the model. The result is F1 score (%).

| $\alpha_2$ | MSL | SMAP | PSM | SWaT | SMD |
|---|---|---|---|---|---|
| 0 | 93.06 | 96.62 | 99.00 | 95.95 | 93.12 |
| 0.1 | 94.10 | 96.35 | 98.63 | 96.57 | 93.70 |
| 0.01 | 93.12 | 96.75 | 98.88 | 96.57 | 94.48 |
| 0.05 | 94.10 | 96.42 | 98.61 | 95.80 | 94.57 |
| 0.005 | 95.45 | 97.21 | 99.12 | 97.09 | 95.45 |
| 0.001 | 93.10 | 96.82 | 98.92 | 95.07 | 94.99 |

## H    VISUALIZATION ANALYSIS

We visualized anomalies and detection results to validate the effectiveness of the proposed H-PAD, as shown in Figure 11. The black dashed line represents the anomaly threshold, with values above this threshold indicating anomalies. The pink highlighted sections denote the true anomaly points. It can be observed that H-PAD assigns higher anomaly scores to anomalies and effectively detects various types of anomalies, including point anomalies, contextual anomalies, and periodic anomalies, further demonstrating the effectiveness of our model.

Table 12: The impact of different sizes of hyperparameters $\alpha_3$ on the model. The result is F1 score (%).

| $\alpha_3$ | MSL | SMAP | PSM | SWaT | SMD |
|---|---|---|---|---|---|
| 0 | 93.41 | 97.11 | 98.94 | 94.17 | 94.44 |
| 0.1 | 92.72 | 96.96 | 98.95 | 92.65 | 94.64 |
| 0.01 | 93.02 | 96.73 | 98.60 | 94.80 | 94.55 |
| 0.001 | 93.19 | 96.57 | 99.09 | 95.94 | 94.34 |
| 0.0001 | 95.45 | 97.21 | 99.12 | 97.09 | 95.45 |

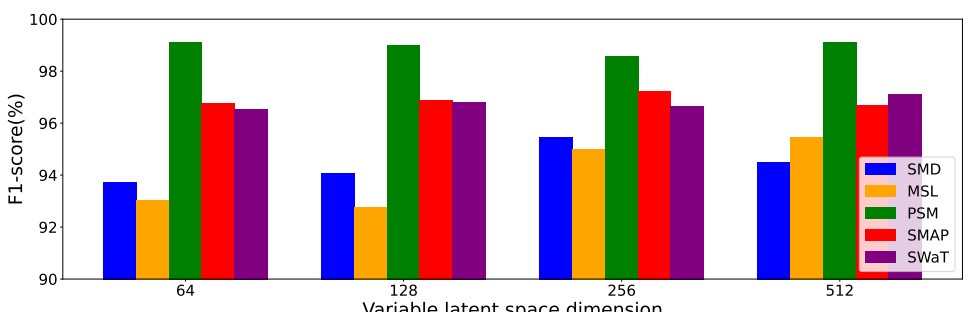

(a) Variable latent space dimension $D$.

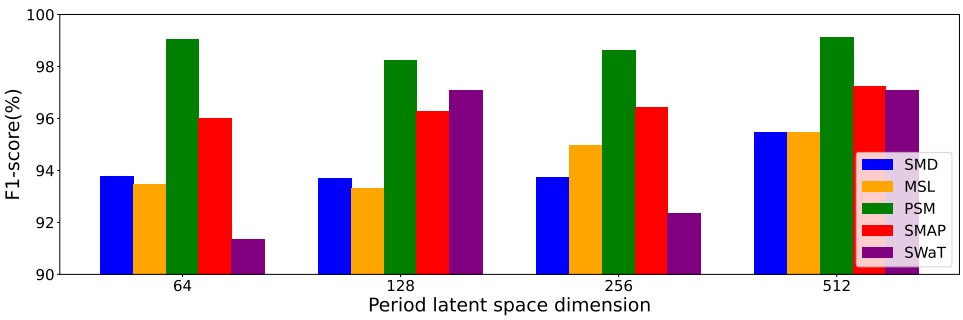

(b) Period latent space dimension $D$.

Figure 10: Hyperparameter sensitivity analysis of the latent space dimension.

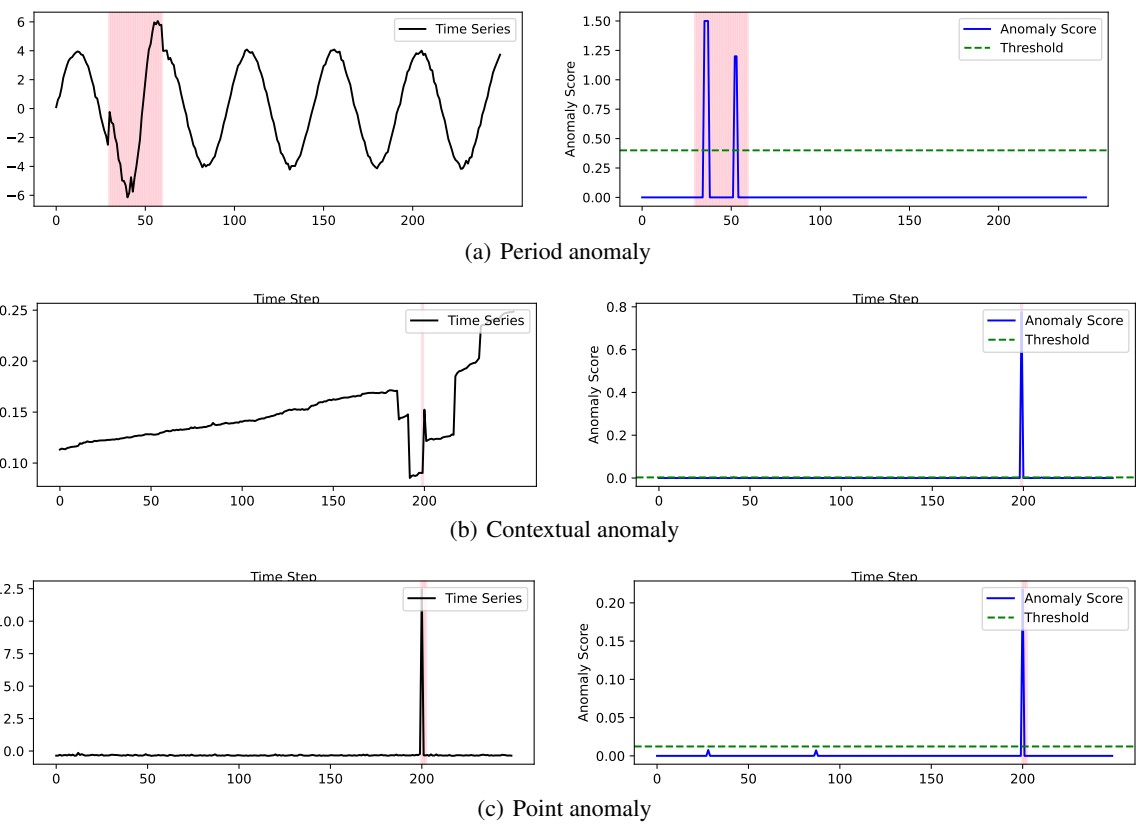

Figure 11: Anomaly visualization (Part 1). For each anomaly, such as periodic anomaly, the left side is the anomaly instance and the right side is the corresponding anomaly score.

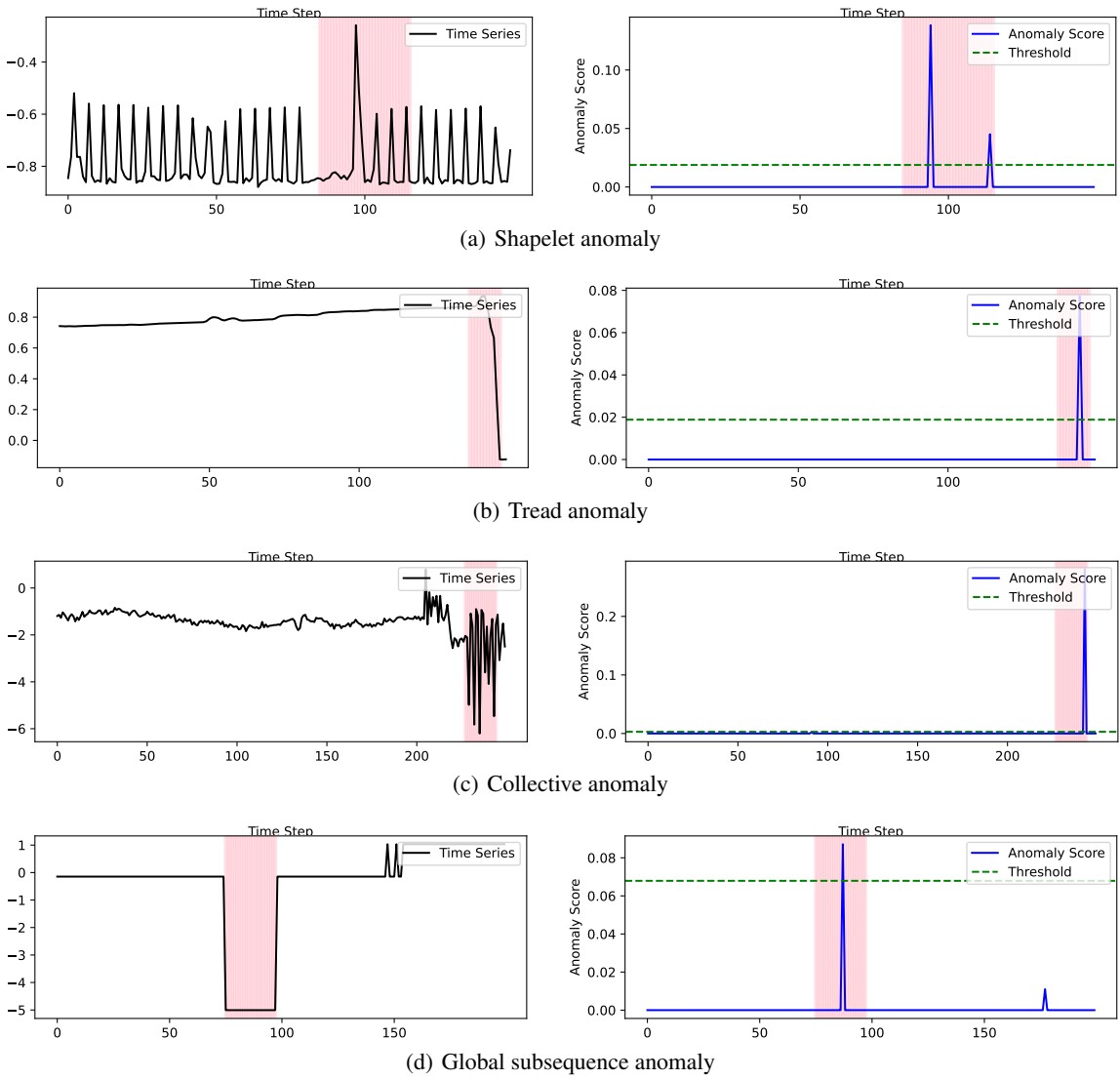

Figure 11: Anomaly visualization (Part 2). For each anomaly, such as periodic anomaly, the left side is the anomaly instance and the right side is the corresponding anomaly score.

