# OpenReview forum: "Learn hybrid prototypes for multivariate time series anomaly detection"
_ICLR.cc/2025/Conference — ICLR 2025 Poster_

### Official Review · Reviewer_HAWs · 2024-10-31

**Soundness:** 2
**Presentation:** 2
**Contribution:** 2
**Rating:** 6
**Confidence:** 4

**Summary:**

This paper proposed a reconstruction-based model called H-PAD for multivariate time series anomaly detection to address the issue of over-generalization.

**Strengths:**

1. Clear motivation
2. Well structured

**Weaknesses:**

1. The font size of the figures is too small.
2. There is a lack of related work, such as "Joint Selective State Space Model and Detrending for Robust Time Series Anomaly Detection".
3. The principle of the proposed method is not clear enough. For example, please explain how the proposed method benefits from mapping the original features to a higher dimensional space (D > C). If C is already very large, will the proposed method still be effective?
4. As shown in Table 1, the performance gain of the proposed method is marginal. Please test it on more datasets say 3 more datasets.
5. Where is your code? the reproducibility is an issue.
6. How did you set the parameters of your proposed method and all compared baselines?
7. How should we choose the parameters of your proposed method?
8. The cases in Figure 5 are overly simple/easy which cannot reflect the advantage of the proposed method.


**I am willing to increase my scores if these issues are well addressed.**

**Questions:**

Please see above.

---

> ### Author Response · Authors · 2024-11-27
>
> We sincerely apologize for our delayed response to your comment. We greatly appreciate the suggestions and questions you have provided. Below, we address each of your points one by one.
> Weakness1:We have adjusted the data size of the figures and tables, but due to page limitations, there may still be some size issues, so please forgive me.
> Weakness2:Thanks for your suggestion, we have added "Joint Selective State Space Model and Detrending for Robust Time Series Anomaly Detection" to the related work and compared it on 7 datasets in Table 2(on page 19).
> Weakness3:We explain the processing of the original data in detail in Appendix B(page 13). First, the original data L×C is mapped to a high-dimensional space, and the C dimension is mapped to the D dimension through an embedding layer (that is, linear projection). The high-dimensional space allows the model to capture more data features and complex patterns. In high-dimensional space, the structure of the data can become more linear or easier to separate, which helps the model to better learn and distinguish features. Moreover, through high-dimensional mapping, the model can more effectively capture the complex nonlinear relationships in the original low-dimensional data. This is especially important for processing complex time series relationships. Normally, the variable dimension C of a time series is generally not very large, and it is common practice to map it to a high-dimensional space. But if C is really large, we think that mapping to a high-dimensional space should also be effective.
> Weakness4:We reevaluated H-PAD using the AUC score as the evaluation metric and introduced two new datasets, NIPS_TS_Water and NIPS_TS_Swan. The results are shown in Table 2(page 9).  Overall, H-PAD achieved strong performance, further demonstrating its effectiveness.
> Weakness5:After we organize the code, we will open source it for reference.
> Weakness6:For the baseline model used for comparison, we used the optimal parameters in the paper code; for the proposed method, we conducted a large number of hyperparameter sensitivity experiments, which can provide a reference for the selection of parameters.
> Weakness7:We have added more hyperparameter sensitivity experiments in Appendix F, which can provide a reference for parameter selection.
> Weakness8:We performed more anomaly visualizations and the results are shown in Figure 9 in Appendix G.

---

> > ### Comment · Reviewer_HAWs · 2024-11-27
> > **thanks for the comments.**
> >
> > I increased my scores. Good luck!

---

### Official Review · Reviewer_WTSw · 2024-11-01

**Soundness:** 2
**Presentation:** 2
**Contribution:** 2
**Rating:** 5
**Confidence:** 4

**Summary:**

This paper proposes a method to address the issue of overfitting to anomalies in existing time series anomaly detection algorithms. The approach involves learning different patches and periodic prototypes, and detecting anomalies through reconstruction. Experiments demonstrate that the proposed method outperforms existing algorithms.

**Strengths:**

Comprehensive experiments were conducted to validate the proposed method.

**Weaknesses:**

1. The paper provides the formulas for the algorithm but lacks an explanation of the rationale and thought process behind their design. This omission may hinder readers' understanding of why the proposed method is effective.
2. The paper primarily claims that current algorithms suffer from overfitting to anomalies. However, subsequent sections on method design do not explain how the proposed method addresses this issue.
3. The text in Figures 1, 3, and 4 is too small.

**Questions:**

1. The paper claims that the proposed method can learn contextual information, and occasional point anomalies cannot utilize this context, thus avoiding overfitting. However, in reality, current time series analysis algorithms can also leverage contextual information. Could the paper provide a clearer explanation of why using multiple patches can mitigate the issue of overfitting to anomalies?
2. In the contributions section, what does “reconstruct abnormal series to be normal ones” mean?
3. Decomposing data using FFT and analyzing time series data from both the time and frequency domains is a common approach in many methods. What are the advantages of the proposed method compared to these existing techniques?

---

> ### Author Response · Authors · 2024-11-27
>
> We sincerely apologize for our delayed response to your comment. We greatly appreciate the suggestions and questions you have provided. Below, we address each of your points one by one.
> Weakness1:We have added more detailed explanations to Average Pooling, Update Patch Prototypes in Section 3.1, and Sparsity Constraints in Section 3.3. In addition, the background in the appendix explains the design ideas in more detail, and the structure of the encoder is described in detail in Appendix B.
> Weakness2:We have introduced the problem of over-generalization in detail in Appendix A(page 13), and hope it will be helpful.
> Weakness3:We are very sorry for the trouble caused to you. We have tried our best to enlarge the symbols in the figure, but due to page limitations, the overall figure is not very large and may still be a little unclear.
> Question1:Although the current model also uses context, due to short-term anomalies, the context information used may contain abnormal information, which may lead to the use of abnormal information to reconstruct the data, thereby reconstructing abnormal data. We explain the problem in detail in Appendix A(page 13). H-PAD uses prototypes of different patches and prototypes of different periods, which can not only suppress point anomalies, but also suppress some short-term anomalies and periodic anomalies. For point anomalies, the difference between point anomalies and normal data is obvious. Only using point normal prototypes can well suppress point anomalies and distinguish abnormal points. For short-term anomalies, short-term anomalies may appear as short-term data fluctuations or sudden changes. Such anomalies may occur in a short period of time, and the normal prototypes of a single point often cannot capture such short-term changes because they need to consider the trends and changes of a series of data points. Therefore, learning normal prototypes of different patch sizes can not only utilize normal local information, but also utilize the trend information of normal patterns, thereby reconstructing short-term anomalies normally. The same is true for periodic anomalies. The normal prototype of a single point usually cannot capture such periodic anomalies because it lacks consideration of the periodic changes in the time series. Therefore, the normal prototype of the period is needed for reconstruction.
> Question2:The normal patch prototypes of different patch sizes can not only reconstruct the normal series into a normal series, but also use the patch prototype to solve the problem of over-generalization and reconstruct the abnormal series into a normal series. With normal patch prototypes of different patch sizes, both normal and abnormal sequences are reconstructed to normal sequences such that the high reconstruction errors for abnormal sequences help the model to detect anomalies.
> Question3:Our advantage is that we use different prototypes to record the normal information of the data. The existing technology reconstructs the data based on the reconstruction model, which often leads to the problem of over-generalization, thus misjudging the anomaly. However, our model uses patch prototypes and period prototypes of different sizes. By solving the problem of over-generalization, the misjudged anomaly is reconstructed into normal through the prototype, resulting in a large reconstruction error for anomaly detection.

---

> ### Author Response · Authors · 2024-11-30
>
> Dear reviewer, hello! We hope that our response and revision addressed your questions and concerns. If you have any further questions or comments, please let us know.

---

### Official Review · Reviewer_u2fE · 2024-11-01

**Soundness:** 2
**Presentation:** 3
**Contribution:** 2
**Rating:** 5
**Confidence:** 4

**Summary:**

The main contribution of this paper lies in proposing a multiscale time series anomaly detection method H-PAD that combines local and periodic information. By designing local and periodic prototypes, introducing sparsity and periodic constraints, and integrating anomaly scoring mechanisms that consider both reconstruction errors and feature space deviations, the method effectively enhances the accuracy and robustness of anomaly detection.

**Strengths:**

1. The paper introduces a framework, H-PAD, for multivariate time series anomaly detection by combining patch-based and period-based prototypes to capture both local and global patterns. Combining local and periodic prototypes offers rich contextual information for anomaly detection.

2. The methodology uses both time-domain and frequency-domain features to enhance detection accuracy. The dual-prototype mechanism, along with tailored anomaly scoring, demonstrates a robust approach to avoiding over-generalization.

3. The reconstruction approach allows the model to effectively replicate normal patterns, aiding in more accurate anomaly identification.

**Weaknesses:**

1. The simple weighted average fusion of local and periodic reconstruction results may lead to information loss or conflict, lacking flexibility.
2. The lack of detailed explanation regarding the implementation mechanism and role of the sparsity constraint may affect understanding and application effectiveness.
3.	The lack of explanation regarding the basis for weight parameters in the loss function may lead to unstable model performance across different tasks.
4.	The setup of the experimental section is not sufficient. Some parameter sensitivity experiments could be conducted to make the theoretical part of the article more convincing.
5.	While the paper is mostly clear, certain aspects, such as some definitions and mechanisms, could benefit from additional clarification to improve replicability and reader comprehension.

**Questions:**

1.	In the INTRODUCTION section of the article, line 65 contains a typographical error: "this paper proposes an MSTAD..." should be "MTSAD."
2.	In line 81, the description of Contribution 2, "but also can reconstruct abnormal series to be normal ones," is not accurately described. Providing a more detailed explanation might be better.
3.	In line 180, it should specify that "z1=1” corresponds to the original sequence X. Adding this detail would be more informative. Additionally, it would be helpful to clarify how the encoder part works—whether it directly uses the encoder block from the Transformer. Providing a more specific structural introduction would improve clarity.
4.	In line 201, the introduction of the update gate is abrupt, and its function is unclear. Additionally, the introduction of the linear transformation matrices U_z and W_p is not well defined—what is their relationship to the context? It would be helpful to explain why linear transformations are applied to b and q.
5.	In section 3.3, line 276, the calculation of the reconstructed sequence involves directly averaging the temporal and frequency domain reconstruction information. Since the sources and characteristics of these two types of reconstruction information are different, is this setting reasonable? It would be advisable to provide some explanation.
6.	In line 296, are alpha_1, alpha_2, and alpha_3 manually adjusted hyperparameters or dynamically learnable parameters using an adaptive method? If they are manually adjusted, how can their approximate ranges be determined? It would be helpful to provide some analyses regarding the parameter settings.

---

> ### Author Response · Authors · 2024-11-27
>
> We sincerely apologize for our delayed response to your comment. We greatly appreciate the suggestions and questions you have provided. Below, we address each of your points one by one.
> Weakness1:Simple weighted averaging for local and periodic reconstruction results can easily lead to information loss or conflict, lacking flexibility. Intuitively, we believe that the information from both sources is equally important, so we employed weighted averaging for fusion. To make the results more convincing, we conducted a parameter sensitivity experiment on the weights, with the results shown in Table 8 of Appendix F.
> Weakness2:Although the normal patterns are learned from normal data, using too many prototypes for reconstruction may occasionally lead to anomalies being reconstructed due to their similarity to normal data. To prevent this, a sparsity constraint is applied, encouraging the model to use fewer prototypes to reconstruct normal features, thereby reducing the likelihood of reconstructing anomalies by chance. Specifically, w represents the weight matrix used for reconstruction with prototypes. By applying an Entropy Loss, the model ensures that a small number of weights approach 1 while the rest approach 0, effectively constraining the model to use fewer prototypes for reconstruction. We conducted a parameter sensitivity analysis on the sparsity constraint weights, as shown in Table 11 of Appendix F(page 18). Setting the weight to 0 results in a performance decline, demonstrating the effectiveness of the sparsity constraint.
> Weakness3:We analyze different values of each weight in Appendix F(page 21). We hope that these analyses can more clearly show the performance of the model on different tasks.
> Weakness4:We have included more hyperparameter sensitivity analyses in Appendix F(page 18), which we hope will provide a clearer picture of the model’s performance.
> Weakness5:Thank you for your comments. We have added more detailed explanations of some definitions and mechanisms in the main text(page 4,5,7) and Appendix A,B(page 13), and hope it will be helpful to readers.For example, the problem of overgeneralization and sparsity constraints are explained in more detail.
> Question1:Thank you for your pointing out the mistakes. We have carefully reviewed the article and corrected some errors.
> Question2:We modified this paragraph to "Patch prototypes can utilize information of different patch sizes. With normal patch prototypes of different patch sizes, both normal and abnormal sequences are reconstructed to normal sequences such that the high reconstruction errors for abnormal sequences help the model to detect anomalies. " We hope it will be easier for readers to understand.
> Question3:Because z1=1, the original time series X remains the original time series after passing through the pooling layer with a pooling kernel size of 1.We have provided clear explanations in the paper to prevent any potential confusion for the readers.We have added the encoder's structural diagram and detailed working principle in Appendix B(page 13), hoping it will be helpful to readers.
> Question4:To obtain prototypes that normal patterns, an update gate a is used to update the prototypes.(on page 5) Since the normal patterns in the prototypes are derived from normal information, we reconstruct the initial prototype b using the similarity matrix v between each prototype b and all normal features q, as well as all the normal features q. The reconstructed prototype vq thus contains normal information. To update the prototypes, the update gate a is employed to determine how much of the original prototype information to retain and how much of the reconstructed prototype information to incorporate. The update gate a is constructed using two linear projections, U and W, applied to the original prototype b and the reconstructed prototype vq, followed by a nonlinear activation function.
> Question5:For the patch prototype, we use the time domain information. For the period prototype, we only use the frequency domain information to get the period, and operate on the period in the time domain information, so the characteristics of the two types are the same. In addition, in order to determine whether direct averaging is effective, we conducted a hyperparameter analysis, as shown in Table 8(on page 19).
> Question6:We have included more hyperparameter sensitivity analyses in Appendix F(page 18), which we hope will provide a clearer picture of the model's performance.

---

> > ### Comment · Reviewer_u2fE · 2024-11-28
> >
> > There is a concern about using point adjustment (PA) for evaluation, which can lead to faulty performance evaluations. Incorporating PA, the Random model outperforms all state-of-the-art models [3].
> >
> > [1] Drift doesn't Matter: Dynamic Decomposition with Diffusion Reconstruction for Unstable Multivariate Time Series Anomaly Detection. NeurIPS 2023.
> >
> > [2] Local Evaluation of Time Series Anomaly Detection Algorithms. KDD 2022.
> >
> > [3] CARLA: Self-supervised contrastive representation learning for time series anomaly detection, arXiv:2308.09296v4, Aug 2024, [Pattern Recognition 157 (2025) 110874]

---

> > > ### Author Response · Authors · 2024-11-28
> > >
> > > It is known that using PA can result in state-of-the-art performance even with random scores or random initialized non-trained models, making it impossible to conduct a fair comparison and assess the effectiveness of the models. To ensure a fair comparison between H-PAD and the baseline models, we used AUC-ROC and AUC-PR as evaluation metrics. As shown in Table 2(page 9), H-PAD achieves the best or second-best results on most datasets. Furthermore, H-PAD exhibited the highest average AUC-ROC score and the second-best AUC-PR score in all seven datasets, highlighting its effectiveness. Among them, AUC-ROC and AUC-PR are the results without point adjustment. In time series anomaly detection, AUC-ROC and AUC-PR are commonly used metrics to evaluate model performance. To ensure that the evaluation is not influenced by the choice of thresholds, these metrics are employed to measure the model's performance across various threshold settings. AUC-ROC (Area Under the Receiver Operating Characteristic Curve) is the area under the ROC curve. The ROC curve is a curve drawn with the false positive rate as the horizontal axis and the true positive rate as the vertical axis. It measures the model's ability to distinguish between positive and negative samples. AUC-PR (Area Under the Precision-Recall Curve) is the area under the Precision-Recall curve. The Precision-Recall curve is a curve drawn with the recall rate as the horizontal axis and the precision rate as the vertical axis. It is more suitable for datasets with imbalanced categories because abnormal samples in anomaly detection often account for a small proportion.

---

> ### Author Response · Authors · 2024-11-30
>
> Dear reviewer, hello! We hope that our response and revision addressed your questions and concerns. If you have any further questions or comments, please let us know.

---

### Official Review · Reviewer_aDfE · 2024-11-02

**Soundness:** 3
**Presentation:** 3
**Contribution:** 3
**Rating:** 6
**Confidence:** 3

**Summary:**

This manuscript proposes a hybrid prototypes learning model, H-PAD, which addresses the problem that existing models can only detect point anomalies. Specifically, normal prototypes are learned from different sizes of patches for time series to discover short-term anomalies. These prototypes in different sizes are integrated together to reconstruct query series so that any anomalies would be smoothed off and high reconstruction errors are produced.  Furthermore, period prototypes are learned to discover periodical anomalies. One period prototype is memorized for one variable of query series.

**Strengths:**

1. The paper is clearly organized.
2. The authors propose H-PAD for multivariate timing anomaly detection, which addresses the problem that existing models can only detect point anomalies.

**Weaknesses:**

1. It is recommended that the authors optimize Fig. 1 to better describe the motivation for this paper.
2. Since anomaly detection is inherently class unbalanced, it is recommended that the authors add AUC to Table 1 to fully analyze the effectiveness of the model.
3. In experiments, whether or not these datasets chosen by the authors contain types of anomalies other than point anomalies seems to be important for the performance of the model. If we only look at Fig. 5, it seems that they are all point anomalies. It is recommended that the authors further add more types of anomalies to the visualization analysis to demonstrate the benefits of H-PAD.

**Questions:**

See Weaknesses please.

---

> ### Author Response · Authors · 2024-11-27
>
> We sincerely apologize for our delayed response to your comment. We greatly appreciate the suggestions and questions you have provided. Below, we address each of your points one by one.
> Weakness1:Time series anomaly detection is an unsupervised task, where normal data is used for reconstruction during the training phase. Since the model is trained on normal data, it learns to reconstruct the time series using normal features. In the testing phase, anomalous data is reconstructed using the normal features learned by the model, which transforms the anomalies into normal patterns. As a result, large reconstruction errors are observed at the anomalous points, allowing anomalies to be identified. However, if the model's reconstruction ability is too strong, it may reconstruct anomalous data as normal, making it difficult to detect anomalies. This is known as the overgeneralization problem.
> By representing the test data with a t-SNE plot, as shown in Figures 5(a)(on page 14) and 5(c), it is evident that the reconstructed data points are very close to the anomalous points of the original data. Due to the overgeneralization problem, it becomes challenging to identify anomalies. To address this, the model learns normal patterns from normal data as prototypes during the training phase. In the testing phase, these learned normal prototypes are used to reconstruct the test data. Since the prototypes only contain normal features, the reconstructed data will exhibit normal characteristics. Finally, to leverage both the normal features and the original features of the test data, the reconstructed normal features are concatenated with the original features and fed into a decoder. The reconstructed normal features suppress the anomalous features, resulting in the final normal reconstructed data.
> H-PAD leverages prototypes of different patches and different periods. This approach not only suppresses point anomalies but also handles short-term and periodic anomalies. For point anomalies, the differences between the anomalies and the normal data are evident, and using only normal point prototypes can effectively suppress point anomalies, enabling anomaly detection. For short-term anomalies, which may manifest as brief data fluctuations or sudden changes over a short period, single-point prototypes often fail to capture such short-term variations as they rely on trends and changes across multiple data points. By learning normal prototypes of varying patch sizes, both the local normal information and the trend information of normal patterns can be utilized, enabling the normal reconstruction of short-term anomalies. The same applies to periodic anomalies; single-point normal prototypes typically cannot capture periodic anomalies due to a lack of consideration for the periodic changes in the time series. Thus, normal prototypes for different periods are required for reconstruction. As shown in Figures 5(b)(on page 14) and 5(d), after reconstruction using different normal prototypes in H-PAD, the reconstructed data is closer to the normal data and farther from the anomalous data. This effectively distinguishes anomalies, allowing the detection of whether the data is anomalous.
>
> Weakness2:We reevaluated H-PAD using the AUC score as the evaluation metric and introduced two new datasets, NIPS_TS_Water and NIPS_TS_Swan. The results are shown in Table 2((on page 9)). Overall, H-PAD achieved strong performance, further demonstrating its effectiveness.
> Weakness3:Period anomaly involves a group of consecutive data points over a specific time range that deviates from the periodic patterns of the time series. These anomalies typically affect multiple continuous time points and are often characterized by irregularities in amplitude or frequency.Therefore, we believe that periodic anomalies include both anomalies in frequency and amplitude, so amplitude anomalies are considered periodic anomalies.In addition, to further verify the effectiveness of H-PAD, we visualized more anomalies. More anomaly visualizations are shown in Appendix G(page 19).

---

> ### Author Response · Authors · 2024-11-30
>
> Dear reviewer, hello! We hope that our response and revision addressed your questions and concerns. If you have any further questions or comments, please let us know.

---

### Official Review · Reviewer_2EhD · 2024-11-03

**Soundness:** 3
**Presentation:** 2
**Contribution:** 2
**Rating:** 6
**Confidence:** 4

**Summary:**

This paper proposes H-PAD, a method to learn hybrid prototypes for multivariate time series anomaly detection. Hybrid prototypes contain both local and global information to help discover both shot-term (point) and long-term (period) anomalies. The authors evaluate their proposed method against various baseline models on 5 datasets and perform ablation studies to understand the importance of each component in the model architecture.

[Update] Adjusted original score after reviewing the authors' rebuttal and revised manuscript

**Strengths:**

1. The authors are familiar with the current literature on time series anomaly detection and evaluate their proposed method against SOTA baselines.
2. Useful ablation studies are performed to understand the importance of each component (patch vs. period prototypes) in the model architecture.
3. The model architecture design using query-based reconstruction (in both temporal and frequency domains) is motivated and explained with clear technical details.

**Weaknesses:**

1. The writing quality should be improved for better clarity. The authors use several non-standard terms such as “different local sizes” which should be corrected.
2. The authors should provide additional implementation details on data processing and model training to help other researchers reproduce and extend their results. For example, how are the patch sizes {z1, z2, …, zm} and k (as in top-k amplitudes of FFT) selected?
3. The authors should discuss the limitations of their work and outline the directions for future research.
4. There are numerous typos and grammatical errors that need to be proofread and corrected. For example, “reference phase” should be “inference phase” (page 1) and “learn and memory prototypes” should be “learn memory prototypes” (page 2).
5. The authors should provide rigorous mathematical definitions of affiliation precision/recall and RAP/RAR since they may not be familiar to most readers. The authors should also clearly explain why these metrics are used instead of the ordinary precision/recall/AUC.
6. It’d be helpful to have more detailed review of the mechanism of memory networks and memory prototypes (either in Related Work or in Supplementary Materials) since these concepts may not be very familiar to most readers.
7. In addition to real-world datasets, it’d be ideal to evaluate the model on simulated time series data to verify that the patch and period prototypes indeed capture multi-scale and multi-period information and effectively detect the corresponding anomalies.

**Questions:**

1. How are the time series data preprocessed? What are the sizes of the datasets? Did the authors apply any filters or normalization to the datasets prior to training the model?
2. How are different types of anomalies (point vs. period) defined in these datasets?
3. What are the computing resources used to train the model? How is the model training efficiency?
4. What are the raw precision, recall and AUC metrics of anomaly detection?
5. How does model performance change with the dimensionality of the time series?
6. What does it mean that “Generally speaking, the series of scale z1 is actually the original series X.”? Does this mean z1 is always set to 1? If so, the authors should clearly state this to avoid confusion.
7. Why is Figure 5 (c) an example of period anomaly instead of point anomaly? It seems that the period is the same but the amplitude is anomalous.
8. What distance metric is used to calculate affiliation precision/recall?

---

> ### Author Response · Authors · 2024-11-28
>
> We sincerely apologize for our delayed response to your comment. We greatly appreciate the suggestions and questions you have provided. Below, we address each of your points one by one.
> Weakness1:Thank you for your suggestions. We have revised the paper and corrected some non-standard terminology.For example, we replaced “different local sizes” with “different patch sizes” to improve clarity and enhance understanding.
> Weakness2:We have added more implementation details to the paper(page 5, 6), including further information on data processing and model training.Additionally, we have included more detailed information about the model in Appendix B for readers' reference. To obtain multi-scale time series, we select patches sequentially from 1 to zm. For example, if scale=5, it means z1=1, z2=2, z3=3, z4=4, z5=5, and different pooling layers are used to obtain time series of five different scales, and time series of different scales are used to learn prototypes of different patch sizes. Regarding the selection of k, since it is a hyperparameter, we performed hyperparameter tuning to achieve the best results. Higher amplitudes contain more significant information. Therefore, we select the top-k amplitudes and derive the corresponding periods based on them. Furthermore, we conducted sensitivity experiments on both the scale and k (i.e., the number of periods) to analyze their impact.
> Weakness3:We have included the limitations of our work and potential directions for future research(page 10). Due to the incorporation of multi-scale information and multi-period information, our model achieves superior results but its training time, number of model parameters, and GPU memory required for training are higher compared to other models. In future work, we plan to optimize the overall framework to improve efficiency, reducing training time and memory consumption without compromising performance. Additionally, we aim to conduct experiments on datasets from a broader range of domains to further validate the robustness of H-PAD.
> Weakness4:Thank you very much for your meticulous review. Indeed, due to time constraints, some spelling and grammatical errors might have been overlooked. We have checked the entire article and corrected any grammatical errors we found.
> Weakness5:Thank you for your suggestion. To help readers become familiar with these evaluation metrics, we have included their mathematical definitions in Appendix D. It is common to use traditional point-based information retrieval measures, such as Precision, Recall, and F1-score, to assess the quality of methods by thresholding the anomaly score to mark each point as an anomaly or not.But mapping discrete labels into continuous data introduces inherent limitations, particularly when evaluating range-based anomalies. While these classical metrics are effective for tasks that assess each sample independently, they fall short for time series datasets, where the temporal dimension is intrinsically continuous. Another notable limitation is the need to define a threshold on the anomaly scores generated by the detection method to classify each time series point as normal or anomalous. However, selecting an appropriate threshold is often challenging and prone to inaccuracies, making it a non-trivial task.In time series anomaly detection, AUC-ROC and AUC-PR are commonly used metrics to evaluate model performance. To ensure that the evaluation is not influenced by the choice of thresholds, these metrics are employed to measure the model's performance across various threshold settings.Afterwards,we use the affiliation metrics, an extension of the classical precision/recall for time series anomaly detection that is local, parameter-free, and applicable generically on both point and range-based anomalies. The metrics leverage measures of duration between ground truth and predictions, and have thus an intuitive interpretation.
> Weakness6:We have included a more detailed review of memory networks and memory prototypes in related work, hoping that it will be helpful to readers.
> Weakness7:In addition to the five real-world datasets, we evaluated the model on two new time series datasets, NIPS_TS_Water and NIPS_TS_Swan. The results are presented in Table 2 of the main text(page 8).Based on comparisons across seven datasets with various baseline models, H-PAD achieves overall optimal performance.

---

> > ### Author Response · Authors · 2024-11-28
> >
> > Question1:1.First of all, the original time series X is divided into multiple subsequences by a sliding window, X={X^1,X^2,...,X^a}. Each subsequence is taken as one time series for training. Subsequently, we input the data from each subsequence into the model for training.(page 4)
> > 2.For the datasets, we provided a detailed description in Appendix C(page 15), including the size of the training and test sets as well as the dimensionality of each dataset.
> > 3.Prior to training the model, we did not apply any filters or normalization to the datasets.
> > Question2:In time series anomaly detection datasets, different types of anomalies are typically defined based on the characteristics and patterns of data points within the sequence. A point anomaly refers to a single data point whose value significantly deviates from its temporal context or usual distribution. Such anomalies are often isolated and do not conform to the local trend or global pattern of the time series. On the other hand, a period anomaly involves a group of consecutive data points over a specific time range that deviates from the periodic patterns of the time series. These anomalies typically affect multiple continuous time points and are often characterized by irregularities in amplitude or frequency.
> > Question3:H-PAD uses the ADAM optimizer with an initial learning rate of 10^(-4). The training process is stopped early within 8 epochs with a batch size of 32. All experiments are implemented in Pytorch using a single NVIDIA GeForce RTX 4090 24GB GPU. The efficiency of the H-PAD training model is compared with another memory model MEDMTO. The results are shown in Appendix E(page 17). Since H-PAD learns patch prototypes of time series of different scales and period prototypes of different periods, its efficiency is much higher than MEMTO.
> > Question4:We provide a detailed introduction in Appendix D(page 16).
> > Precision: The proportion of data points predicted to be abnormal that are actually abnormal is calculated as follows:precision=(TP)/(TP+FP).
> > Recall: The proportion of data points that are correctly predicted to be abnormal among the data points that are actually abnormal is:recall=(TP)/(TP+FN}).
> > AUC-ROC (Area Under the Receiver Operating Characteristic Curve) is the area under the ROC curve. The ROC curve is a curve drawn with the false positive rate (FPR) as the horizontal axis and the true positive rate (TPR) as the vertical axis. It measures the model's ability to distinguish between positive and negative samples.
> > AUC-PR (Area Under the Precision-Recall Curve) is the area under the Precision-Recall curve. The Precision-Recall curve is a curve drawn with the recall rate (Recall) as the horizontal axis and the precision rate (Precision) as the vertical axis. It is more suitable for datasets with imbalanced categories because abnormal samples in anomaly detection often account for a small proportion.
> > Question5:We have added several new hyperparameter sensitivity experiments in Appendix F(page 18), including using different latent space dimensions D in five datasets. It evaluates the impact of changes in feature dimensions D on model performance.
> > Question6:Because z1=1, the original time series X remains the original time series after passing through the pooling layer with a pooling kernel size of 1.We have provided clear explanations in the paper(page 5) to prevent any potential confusion for the readers.
> > Question7:Period anomaly involves a group of consecutive data points over a specific time range that deviates from the periodic patterns of the time series. These anomalies typically affect multiple continuous time points and are often characterized by irregularities in amplitude or frequency.Therefore, we believe that periodic anomalies include both anomalies in frequency and amplitude, so amplitude anomalies are considered periodic anomalies.In addition, to further verify the effectiveness of H-PAD, we visualized more anomalies. More anomaly visualizations are shown in Appendix G(page 19).
> > Question8:We adopted the evaluation criteria proposed in "Local Evaluation of Time Series Anomaly Detection Algorithms". The distance metric used for calculating affiliation precision/recall is the average distance between sets, to measure how far the events are one from each other.On one hand, if most anomalies in the predicted set significantly overlap with the ground truth anomalies, the average directed distance from the predicted set to the ground truth set will be smaller, resulting in a higher Aff-P for the model. On the other hand, if most anomalies in the ground truth set are covered by the predicted set, the average directed distance from the ground truth set to the predicted set will also be shorter, leading to a higher Aff-R for the model.

---

> ### Author Response · Authors · 2024-11-30
>
> Dear reviewer, hello! We hope that our response and revision addressed your questions and concerns. If you have any further questions or comments, please let us know.

---

> > ### Comment · Reviewer_2EhD · 2024-12-01
> > **Adjusted original score and left new comments**
> >
> > I thank the authors for carefully addressing the reviewers' comments, providing additional results, and improving the quality of the paper, although the rebuttal was submitted after the public discussion period (which ended on Nov 26). After reviewing the authors' rebuttal and revised manuscript, I adjusted my score accordingly.
> >
> > Please find my additional comments below:
> >
> > 1. The results in Table 2 show that the proposed method H-PAD has inferior performance in AR and AP on 4 of the 8 benchmark datasets. Further more, H-PAD shows a large gap in lower AR on 3 benchmark datasets (SMAP, SMD, NIPS_TS_Water). This indicates that the H-PAD method is not yet optimal and cannot be claimed as the "state-of-the-art performance" as the authors stated in the the Abstract.
> >
> > 2. Since the training is done by dividing the original time series into sliding windows (or subsequences), the author should explain how they split the training and test datasets in Table 5. In particular, why is some dataset equally split (e.g., SMD, NIPS) while SMAP contains much more data in test than training data?
> >
> > 3. The comparison of F1 scores is difficult to see in Fig 4 and Fig 8 since all F1 scores are above 0.8. It'd be better to change the y-axis range to [80, 100] instead of [0, 100] to clearly illustrate the difference.

---

> > > ### Author Response · Authors · 2024-12-02
> > >
> > > Dear reviewer, thank you very much for your comments!
> > > 1. Indeed, the performance of our AUC-ROC and AUC-PR indicators is not as good as the F1 score after point adjustment. But the effects of other baseline models are also uneven. They can achieve a good effect on individual datasets, and the overall effect may not be so ideal. However, overall, our model can achieve the best or sub-optimal effect on more datasets, and our average AUC-ROC and AUC-PR can also achieve a good effect. We think this can also explain some of the advantages of our model. But indeed, as you said, our model does not achieve the best on all datasets. If possible, we will modify the  Abstract. Thank you very much for your suggestions.
> > > 2. I am sorry for the division of training and test datasets. These datasets are commonly used datasets for time series anomaly detection. These datasets are published by others, and we only use them, so we have not carefully understood the division of training and test sets in the datasets. I am sorry for troubling you.
> > > 3. Thank you for your suggestions, but due to time constraints, we cannot make modifications. We will revise your suggestion if we have the chance.

---

### Meta-Review · Area_Chair_iFJa · 2024-12-17

**Metareview:**

This work proposes H-PAD, a hybrid prototypes model aiming to handle both short-term (point) and periodical anomalies in multivariate time series by reconstructing normal data patterns and identifying anomalies via reconstruction errors.

The authors were responsive (I personally appreciate this effort!), adding clarifications, new experiments, and tuning their explanations after the initial round of comments. They introduced more details on data preprocessing, parameter choices, and included additional datasets and metrics (AUC, PR) following the reviewers’ requests. They also tried addressing concerns on methodology clarity and provided some ablation and hyperparameter analyses.

After the rebuttal, reviewers showed some improvement in scores or understanding. Still, not all concerns were fully resolved, as at least one reviewer was still not fully convinced about the comparison fairness and clarity, while another requested more datasets and a more thorough explanation. Another reviewer, though, found the improvements and added details satisfactory enough to raise their score.

Common threads mentioned by multiple reviewers include the need for clearer explanations of design choices (especially how prototypes and memory constraints work), more rigorous comparisons against other methods, and better clarity around parameter selection and evaluation metrics. The authors did attempt to fix these, while I believe it is fair to leave this for camera ready if accepted.

**Additional Comments On Reviewer Discussion:**

see above

---

### Decision · Program_Chairs · 2025-01-22

Accept (Poster)